



# Impact of transport model resolution and a-priori assumptions on inverse modeling of Swiss F-gases emissions

Ioannis Katharopoulos[1,2], Dominique Rust[1,3], Martin K. Vollmer[1], Dominik Brunner[1,2], Stefan Reimann[1], Simon J. O'Doherty[4], Dickon Young[4], Kieran M. Stanley[4,5], Tanja Schuck[5], Jgor Arduini[6,7], Lukas Emmenegger[1], and Stephan Henne[1]

[1]Laboratory for Air Pollution / Environmental Technology, Empa, Swiss Federal Laboratories for Materials Science and Technology, Dübendorf, Switzerland
[2]Institute for Atmospheric and Climate Science, ETH Zurich, Zurich, Switzerland
[3]Department of Chemistry and Applied Biosciences, ETH Zurich, Zurich, Switzerland
[4]School of Chemistry, University of Bristol, Bristol, United Kingdom
[5]Institute for Atmospheric and Environmental Science, Goethe University Frankfurt, Frankfurt am Main, Germany
[6]Department of Pure and Applied Sciences, University of Urbino, Urbino, Italy
[7]CNR-ISAC, National Research Council of Italy, Institute of Atmospheric Sciences and Climate, Bologna, Italy

**Correspondence:** Ioannis Katharopoulos (ioannis.katharopoulos@gmail.com), Stephan Henne
(stephan.henne@empa.ch)

**Abstract.** Inverse modeling is a widely used top-down method to infer greenhouse gas (GHG) emissions and their spatial distribution based on atmospheric observations. The errors associated with inverse modeling have multiple sources, such as observations and a-priori emission estimates, but they are often dominated by the transport model error. Here, we utilize the Lagrangian Particle Dispersion Model (LPDM) FLEXPART, driven by the meteorological fields of the regional numerical weather prediction model COSMO. The main source of errors in LPDMs is the turbulence diffusion parameterization and the meteorological fields. The latter are outputs of an Eulerian model. Recently, we introduced an improved parameterization scheme of the turbulence diffusion in FLEXPART, which significantly improves FLEXPART-COSMO simulations at 1 km resolution. We exploit F-gases measurements from two extended field campaigns on the Swiss Plateau (in Beromünster and Sottens) and we conduct both high- (1 km) and low-resolution (7 km) FLEXPART transport simulations that are then used in a Bayesian analytical inversion to estimate spatial emission distributions. Our results for four F-gases (HFC-134a, HFC-125, HFC-32, $SF_6$) indicate that both high-resolution inversions and a dense measurement network significantly improve the ability to estimate spatial distribution of the emissions. Furthermore, the total emission estimates from the high-resolution inversions ($351\pm44\,\mathrm{Mg\,yr^{-1}}$ for HFC-134a, $101\pm21\,\mathrm{Mg\,yr^{-1}}$ for HFC-125, $50\pm8\,\mathrm{Mg\,yr^{-1}}$ for HFC-32, $9.0\pm1.1\,\mathrm{Mg\,yr^{-1}}$ for $SF_6$) are significantly higher compared to the low-resolution inversions (20-40 % increase) and result in total a-posteriori emission estimates that are closer to national inventory values as reported to the UNFCCC (10-20 % difference between high-resolution inversion estimates and inventory values compared to 30-40 % difference between the low-resolution inversion estimates and inventory values). Specifically, we attribute these improvements to a better representation of the atmospheric flow in complex terrain in the high-resolution model, partly induced by the more realistic topography. We further conduct numerous sensitivity inversions, varying different parameters and variables of our





Bayesian inversion framework to explore the whole range of uncertainty in the inversion errors (e.g., inversion grid, spatial distribution of a-priori emissions, covariance parameters like baseline uncertainty and spatial correlation length, temporal resolution of the assimilated observations, observation network, seasonality of emissions). From the above-mentioned parameters, we find that the uncertainty of the mole fraction baseline and the spatial distribution of the a-priori emissions have the largest impact on the a-posteriori total emission estimates and their spatial distribution. This study is a step to-

wards mitigating the errors associated with the transport models and better characterizing the uncertainty inherent in the inversion error. Improvements in the latter will facilitate the validation and standardization of the national GHG emission inventories and support policymakers.

## 1 Introduction

Monitoring greenhouse gas (GHG) emissions into the atmosphere is critical in order to justify whether they conform to our
endeavor of limiting the average global temperature increase below 2°C from pre-industrial levels. Bottom-up methods quantify GHG emissions from statistical data, by employing activity data and emission factors for the relevant emission processes, without cross-validating the results against actual atmospheric observations (Leip et al., 2018). Depending on the emitting process, bottom-up methods may be afflicted by large uncertainties, especially when spatially-resolved emissions are considered on sub-national scales, and when the emitting processes are not well understood or are more complex
than what can be described through an emission factor approach or emission process models (Leip et al., 2018). Top-down methods employ atmospheric observations to infer the total surface fluxes and their spatial distribution. With the increasing observational network coverage, high-resolution satellite observations, and the increasing accuracy of atmospheric transport models, top-down methods have become a powerful tool for estimating the emissions of GHGs and validating bottom-up inventories from the global to the local scale (Nisbet and Weiss, 2010; Weiss and Prinn, 2011; Leip et al., 2018;
Jacob et al., 2022).

Atmospheric inverse modeling is a widely applied top-down emission estimation method (Bergamaschi et al., 2018), combining observations of atmospheric compounds, atmospheric transport models, a-priori estimates of the surface emission fluxes, and inversion frameworks to deduce the most likely state of the surface-emission fluxes for the compound of interest. Atmospheric transport models are utilized in inversions to link the tracer's sources and the observed mole frac-
tions at a receptor. These models advect and disperse the tracer from the source to the receptor and are either based on or part of numerical weather prediction (NWP) models, such as COSMO-GHG (Jähn et al., 2020) and WRF-chem (Grell et al., 2005), or on Lagrangian particle dispersion models (LPDMs) offline coupled with NWP models (NWP meteorological fields drive LPDMs), such as FLEXible PARTicle Dispersion Model (FLEXPART, Stohl et al., 2005) (used in this study), Stochastic Time-Inverted Lagrangian Transport (STILT, Lin et al., 2003), Hybrid Single Particle Lagrangian Integrated Trajectory (HYS-
PLIT, Stein et al., 2015), Numerical Atmospheric-dispersion Modelling Environment (NAME, Jones et al., 2007), and others. The big advantage of LPDMs over Eulerian models is their straightforward applicability in both forward and backward in-time simulations (Seibert and Frank, 2004; Thomson and Wilson, 2012), with the backward mode allowing the direct





calculation of source-receptor relationships (Seibert and Frank, 2004). Source-receptor relationships provide the influence of an emission source on the observed values at the receptor site. Thus, they provide a direct link between mole fractions and emissions, required for inverse modeling, something that is harder to deduce in Eulerian models.

Errors in the inverse modeling estimates are introduced by errors in the atmospheric observations, in the estimate of the a-priori, and in the error covariance matrices, but are strongly driven by errors due to transport and representativeness inherent in the transport model (Bergamaschi et al., 2018; Karion et al., 2019). The main sources of transport model errors are the representation of boundary layer dynamics, vertical mixing, and the horizontal and vertical resolution of the models (Karion et al., 2019). Some of these errors come from the NWP models driving the LPDMs, and others, from the LPDMs themselves. Hence, differences in the simulated mole fractions at the receptor sites can occur either when different NWP models are used to drive the LPDM or when different LPDMs are used for the advection and the dispersion of the tracers. Along with increasing transport model resolution, the model topography converges to the real topography, leading to a better representation of terrain-induced flow, especially in complex terrain (e.g., Schmidli et al., 2018). Thus, transitioning from low- to high-resolution NWP models to drive the LPDMs should directly reduce the representation and transport model errors.

The inversions conducted in this study focus on some of the most important (by $CO_2$-equivalent emissions) synthetic GHGs released in Switzerland (three hydrofluorocarbons (HFCs) and $SF_6$). HFCs are not directly part of the energy system (like $CO_2$ from fossil fuel use) nor the agricultural system (like $CH_4$ and $N_2O$), since their emissions only stem from direct anthropogenic production and usage. Even if they do not play a significant role in the energy system now, they are used in heat pumps and air conditioners. Thus, a potential decline of fossil fuels will possibly see increased emissions from refrigerants. Hence, we should tackle them with low-GWP alternatives to decrease their impact. HFCs were introduced to replace chlorine and bromine-containing ozone-depleting substances. The latter have been regulated by the Montreal Protocol, which was very successful in preventing further damage to the ozone layer (Engel et al., 2018). Next to their role as refrigerants, HFCs are utilized on a large scale as foam blowing agents, aerosol propellants, solvents and as fire suppressants. HFCs do not deplete the stratospheric ozone layer, but some of them have very significant GWP of up to 14,000 on a 100-year perspective. Their abundance in the atmosphere has been continuously increasing due to their widespread usage (Velders et al., 2022), and if their emissions were left uncontrolled, their impact on global surface warming would be, according to projections, 0.3–0.5 °C by the end of the century (Velders et al., 2022). The members of the Montreal Protocol agreed through the Kigali amendment in 2016 to regulate the emissions of HFCs and gradually reduce their emissions and phase down the substances with the highest GWPs by 2040. Bottom-up estimates of synthetic GHG emissions are connected to large uncertainties in the leakage rates of these compounds from various applications (e.g., refrigeration, foam blowing). Thus, continuous atmospheric monitoring and top-down emission estimation is necessary to validate the bottom-up national inventories and assess whether the GHG emissions are in line with the new regulations now in effect in most developed countries (Velders et al., 2022).

In this study, we use the LPDM FLEXPART-COSMO (Henne et al., 2016; Pisso et al., 2019), driven by operational meteorological analysis fields created by MeteoSwiss with the regional NWP model COSMO. The main focus of this study is the





comparison of inversions using COSMO at two different spatial resolutions (7 km × 7 km and 1 km × 1 km). In previous studies (Henne et al., 2016; Bergamaschi et al., in review, 2017) FLEXPART-COSMO was successfully operated at 7 km × 7 km spatial resolution. Recently, we introduced a new turbulence scheme for FLEXPART-COSMO (Katharopoulos et al., 2022), which makes high resolution, 1 km × 1 km, FLEXPART-COSMO simulations more realistic. Operating FLEXPART-COSMO-1 with FLEXPART's default turbulence scheme leads to an overestimation of turbulence and, hence, excessive tracer dispersion. Applied to methane observations in Switzerland, FLEXPART-COSMO-1 with the new turbulence scheme outperforms the low-resolution FLEXPART-COSMO-7 by producing more realistic peak concentration amplitudes and correlation with the observations (Katharopoulos et al., 2022).

Newly available synthetic gas observations, collected as part of the Swiss National Science Foundation (SNSF) project IHALOME (Innovation in Halocarbon Measurements and Emission Validation), from the Swiss Plateau at the Beromünster and Sottens tall towers, complemented with observations from the Advanced Global Atmospheric Gases Experiment (AGAGE) network (Prinn et al., 2018), allow us to localize and quantify the emissions in Switzerland and in neighboring countries. Before IHALOME, F-gas emissions in Switzerland had to be inferred from measurements at the Jungfraujoch station, which has a comparatively low sensitivity to emissions over the Swiss Plateau due to its remote location and high altitude in the Swiss Alps (FOEN, 2022). F-gases measurements from AGAGE sites have been repeatedly used in the past for inverse modeling studies to estimate European emissions on the continental and/or national scale (e.g., Manning et al., 2003; Stohl et al., 2010; Brunner et al., 2012; Ganesan et al., 2014; Lunt et al., 2015; Brunner et al., 2017; Rust et al., 2022; Manning et al., 2021). In Rust et al. (2022) the observations from only Beromünster combined with low-resolution (7 km) transport simulations were used to infer the total Swiss emissions for 29 halocarbons. Here, we utilize both high- and low-resolution transport simulations and observations from both campaigns in Beromünster and Sottens. The first question that our study assesses is whether the high-resolution simulations can enhance the capability of the inversion method to localize emissions. The second question is whether the combination of high-resolution inverse modeling with a denser measurement network further helps in the estimation and localization of emissions on the national scale.

The inversion system employed in this study is an analytical Bayesian inversion system (Brasseur and Jacob, 2017) coupled with a maximum likelihood optimization method (Michalak et al., 2005) in order to obtain objective estimates for the parameters of the covariance matrices. This method was shown to underestimate the uncertainty of the emissions (e.g., Berchet et al., 2015) due to the Gaussian errors assumption. To explore how different inversion setups impact the national total a-posteriori emissions and their spatial distribution and uncertainty, we further conducted a series of sensitivity inversions where we varied different parameters and aspects of our inversion problem.

The manuscript is organized as follows: Sect. 2 describes the observational sites and measurements, the different versions of FLEXPART utilizing inputs from different NWPs, the inversion framework, and the different sensitivity inversions conducted to explore the range of uncertainty for the posterior state vector. In Sect. 3 we present the inversion results for the main HFCs and $SF_6$ for the different model resolutions, different combinations of observational data and additional sensitivity inversions. Finally, in Sect. 4 we discuss our findings and conclusions.



## 2 Methods

### 2.1 Measurement sites

The details of the observational sites used, such as their coordinates, their altitude, the air inlet height above ground, and
125 the height of each site in the different transport model versions (Sect. 2.4), are summarized in Table 1. Their location can
be seen in Fig. 1. Since the main goal of this study is to quantify the differences between low- (7 km) and high-resolution
inversions (1 km) in Switzerland, the observational sites chosen should be sensitive to Swiss emissions. Most of the Swiss
F-gas emissions can be expected to originate from the region called the Swiss Plateau. It is located north of the Alps, cov-
ering about 1/3 of the area of Switzerland, and including about 2/3 of the population of Switzerland. The biggest cities of
130 Switzerland are located in this region and most of the industrial activity takes place here as well.

The Beromünster (BRM) tall tower site (Table 1) is located in the middle of the Swiss Plateau on a hill with an elevation
of about 800 m a.s.l.. GHG measurements at the tower were established in 2012 (Berhanu et al., 2015; Oney et al., 2015)
and since 2016 the site is part of the Swiss air quality observing network (NABEL). The area surrounding Beromünster is
mainly rural, and used for agricultural activities. BRM is sensitive to emissions from most of the Swiss Plateau, as can be
seen in Fig. 2. The closest city to Beromünster is Lucerne (urban area population of 220 000), located 20 km south of the
site, whereas the Zurich urban area (approximately 1.3 million) is 40 km to the east. The measurements on the site were
taken at a tall tower with height 217 m a.g.l., at a sample inlet height of 212 m a.g.l.

The Sottens (SOT, Table 1) tall tower is located in the Western part of the Swiss Plateau in the canton of Vaud at an altitude
of about 800 m a.s.l. The area surrounding SOT is also rural. Measurements in SOT are sensitive to emissions from the west
and center parts of the Swiss Plateau as well as from the deep Alpine Rhone valley (canton of Valais, Fig. 2). The closest
larger city is Lausanne (urban area population of 430 000), located 20 km south-west of the site, while Geneva is about
80 km west south-west of the site (urban area population of 500 000). The measurements on the tall tower were taken at an
inlet height of 120 m a.g.l.

The Jungfraujoch (JFJ, Table 1) site is a high-altitude observatory located in the Bernese Alps on the boundary between
145 the cantons of Valais and Bern. The observatory is located at a steep mountain saddle connecting two major mountains
(Jungfrau, 4158 m a.s.l. and Mönch, 4099 m a.s.l.). JFJ is part of the AGAGE network and has been measuring halocarbons
since 2000. Although JFJ is representative of lower free tropospheric conditions in the winter, it frequently receives fresh
boundary layer pollution during the summer months, both from the Swiss Plateau and from the South of the Alps, but also
from more distant sources throughout Central Europe (Henne et al., 2010; Herrmann et al., 2015).

Two additional sites were used in the inversions conducted within this study: Mace Head (MHD) and Tacolneston (TAC),
Table 1. MHD is located in County Galway on the west coast of Ireland. Its exposure to the North Atlantic Ocean makes it
an ideal location for background observations due to the dominating westerly flow. TAC is located 150 km to the northeast
of London on east coast of England in south Norfolk. Its location is optimal to constrain emissions from the UK and partly
from the Benelux region, which is one of the regions with the highest emission density in Europe (Manning et al., 2021).
These sites were also used in a previous study to constrain Swiss halocarbon emissions (Rust et al., 2022). Adding these





**Table 1.** Details of the observational sites used in the study, including location, altitude, and the height of the model topography in the different FLEXPART model versions.

| Station | ID | Longitude (°E) | Latitude (°N) | Altitude (m a.s.l.) | COSMO-7 height (m a.s.l.) | COSMO-1 height (m a.s.l.) | IFS height (m a.s.l.) | Inlet height (m) |
|---|---|---|---|---|---|---|---|---|
| Beromünster | BRM | 8.1755 | 47.1896 | 797 | 615 | 718 | - | 212 |
| Sottens | SOT | 6.7364 | 46.6559 | 776 | 718 | 764 | - | 120 |
| Jungfraujoch | JFJ | 7.9851 | 46.5475 | 3580 | 2653 | 3354 | - | 2 |
| Tacolneston | TAC | 1.1386 | 52.5177 | 56 | - | - | 44 | 185 |
| MaceHead | MHD | -9.8995 | 53.3258 | 8 | - | - | 8 | 2 |
| Taunus Observatory | TOB | 8.4473 | 50.2225 | 825 | 427 | 517 | - | 8 |
| Monte Cimone | CMN | 10.7007 | 44.1935 | 2165 | 1228 | 1774 | - | 13 |

sites outside Switzerland allows the inversion to constrain larger-scale European emissions. Leaving these unconstrained by observations may have led to biased estimates for the Swiss domain.

Two more AGAGE sites were employed in sensitivity inversions to explore any further impact of additional observations on Swiss emissions: Monte Cimone (CMN) and Taunus Observatory (TOB), Table 1. CMN is a high-altitude observatory on the highest peak of the Northern Apennines in Italy. Its remote location, high altitude, and large distance from big cities and hence major emission sources, make it representative of the free troposphere and background values in South Europe and the North Mediterranean basin, but it can also occasionally receive pollution events from the Po Valley (Bonasoni et al., 2000). TOB is located on the second highest peak in the Taunus mountain region in central Germany. Its close proximity to major emission sources (Frankfurt and Mainz), and its location in central Europe make the site well suited for European air pollution studies.

## 2.2 Observational data

In this study, we used observational data of 1,1,1,2-tetrafluoroethane (HFC-134a), 1,1,1,2,2-pentafluoroethane (HFC-125), difluoromethane (HFC-32), and sulfur hexafluoride ($SF_6$), which together with 1,1,1-trifluoroethane (HFC-143a, not reliably measured from BRM and SOT) account for more than 80% of total Swiss halocarbon emissions in terms of $CO_2$-equivalents (Reimann et al., 2021). The observational data come from two extended measurement campaigns at BRM and SOT, conducted within the project IHALOME (Rust et al., 2022), and from the AGAGE monitoring network (Prinn et al., 2018). The measurement campaigns at BRM and SOT were performed to explore the impact of a denser measurement network on the Swiss national emission estimates by top-down methods. Semi-continuous air samples were taken with a frequency of approximately two ambient air measurements within three hours using a Medusa pre-concentration unit, coupled to gas chromatography and mass spectrometry (Miller et al., 2008). In total, about 60 fully calibrated halocarbons were measured with atmospheric abundances in the dry-air mole fraction range of parts-per-quadrillion (femtomol $mol^{-1}$)





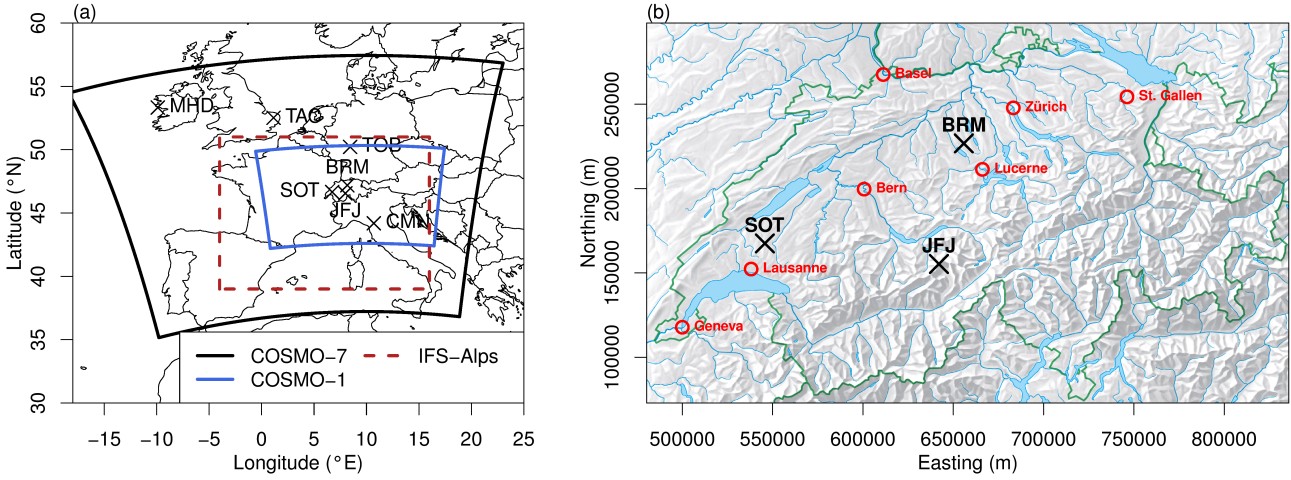

**Figure 1.** Measurement locations (black crosses) of sites used in the inversions and COSMO and IFS model domains (polygons) as used as input to FLEXPART (a). Swiss measurement locations (black crosses) and major cities (red circles) on top of topographic relief including major rivers and lakes (Swiss coordinate system, LV03) (b).

to parts-per-trillion (picomol mol$^{-1}$ (Rust et al., 2022)). Measurements of HFC-143a suffered from instrumental problems and were therefore not used in the present analysis. Measurements at MHD and TAC were also performed with a Medusa-GCM instrument, while those at TOB and CMN with different preconcentration-GC/MS instruments (Schuck et al., 2018;

Maione et al., 2013). The measurements used in the present study are based on fully intercalibrated reference standards. The measurements are based on the Scripps Institution of Oceanography (SIO) primary calibration scales SIO-05 for HFC-134a and $SF_6$, SIO-14 for HFC-125, and SIO-07 for HFC-32.

The observational data employed for the inversions cover the period from August 2019 to October 2021. Data from TAC, MHD, and JFJ were used for the whole period, while BRM and SOT data were available only during the field campaigns.

The campaign in BRM lasted from August 2019 to September 2020, and in SOT from March 2021 to October 2021. There was no temporal overlap because the same instrument had to be used at both locations. CMN and TOB observations were employed in sensitivity inversions for the BRM campaign period only. Measurements from TOB come from flask samples, which are collected weekly for offline analysis.

### 2.3    Baseline

To run our inversions, 24-hourly (3-hourly for sensitivity inversion) aggregates were produced from the available observations of the above-mentioned sites. To correctly infer regional emissions from a limited model domain, accurate knowledge of the so-called background (or baseline) mole fraction of a compound is needed. An observed mole fraction of a compound can be decomposed into a baseline fraction, $y_{o,b}$, and the contribution due to recent emissions, as targeted by



**Figure 2.** Simulated total surface sensitivity (footprints) for Beromünster, (a) and (b), and for Sottens, (c) and (d), for the duration of the measurement campaigns (01 09 2019–31 08 2020 for BRM and 05 03 2021–24 10 2021 for SOT) as obtained from the FLEXPART-COSMO-7, (a) and (c), and FLEXPART-COSMO-1, (b) and (d). The surface sensitivity is given as particle residence time per air density. The locations of the sites are indicated with a black crosses, major cities with black circles.



the regional simulation, $y_{o,p}$

$$y_o = y_{o,b} + y_{o,p}. \tag{1}$$

An underestimation of the baseline will magnify an emission event, whereas an overestimation will reduce the intensity of an emission event. We estimate our baseline mole fractions by using the robust extraction of the baseline signal (REBS) method (Ruckstuhl et al., 2012). The REBS method is an iterative filter, which assumes that the mean of the baseline can be approximated by a smooth curve and its uncertainty distribution can be given by a Gaussian distribution with a con-

stant (in time) standard deviation. The smooth curve is estimated by piece-wise local weighted linear regressions. The first weight function acts to decrease the impact of observations in proportion to their distance from the point of the time series that is to be estimated, $t_0$. The second reduces the influence of the data according to their distance from the expected value of the baseline. In each iteration, an updated estimate of the baseline value and the baseline standard deviation is calculated, and the method usually converges after 5-10 iterations. For our baseline estimates, a tuning factor of b = 3.5,

a temporal window width of 60 days, and a maximum of 10 iterations were used. In our inversions, we used the baseline estimated from JFJ for all Swiss sites, the one estimated for MHD for the sites on the British Isles and TOB, and the one estimated for CMN for CMN itself. This selection is motivated by the fact that the REBS method works best for sites that are mostly sampling background, whereas for typical continental boundary layer sites (like BRM, SOT, TAC) only a few 'pure' background observations exist throughout the year and, hence, REBS-estimated baselines tend to overestimate true

baselines. All baselines were updated as part of the emission inversion step, individually for each site.

## 2.4 Transport models

The inversion system utilized for this study is comprised of an atmospheric transport model, which relates the spatial emissions, $\mathbf{x}$, of the compound of interest to the mole fractions measured at the receptor site, $\mathbf{y_o}$, via a linear mapping $\mathbf{y_o} = \mathscr{H}\mathbf{x}$. Here, the LPDM FLEXPART (Pisso et al., 2019) was driven by the meteorological fields from two Eulerian NWP

models: the limited-area NWP model COSMO, and the Integrated Forecasting System (IFS) of the European Centre for Medium-range Weather Forecasts (ECMWF). Simulations with IFS were used to extend FLEXPART-COSMO simulations beyond the COSMO model domain (Katharopoulos et al., 2022).

### 2.4.1 COSMO & IFS models

COSMO is a non-hydrostatic limited-area atmospheric model. It was initially designed for operational NWP by the German

weather service (DWD) and it is still used by several national weather services including MeteoSwiss (Baldauf et al., 2011). Its final version was released on December 15, 2021, while a transition to the ICON (ICOsahedral Nonhydrostatic) model is considered for most of the meteorological services using COSMO, including MeteoSwiss (envisaged for 2023). MeteoSwiss has been operating COSMO at three different spatial resolutions: COSMO-7 with a grid spacing of 6.6 km (from 2002-02-01 to 2020-10-29), COSMO-2 with a grid spacing of 2.2 km (from 2008-02-19 to 2023) and COSMO-1 with a grid spacing of 1.1

km (from 2015-09-30 to present) (Schmidli et al., 2018; Klasa et al., 2018; Leuenberger et al., 2020). The domain for the low-





and the high-resolution model versions can be seen in Fig. 1. The low-resolution model domain covers parts of central and western Europe (-10° to 20° E and 38° to 55° N; Fig. 1). The higher-resolution operational domain of MeteoSwiss COSMO-1 focuses on Switzerland and the Alps and has a considerably smaller extent (approximately from 0° to 17° E and 43° to 50° N; Fig. 1). Operational COSMO is driven by initial and boundary conditions from ECMWF IFS. COSMO analysis fields are

available from MeteoSwiss at a temporal resolution of 1 hour at all spatial resolutions mentioned above.

High resolution (HRES) IFS is the operational global NWP model of ECMWF. The HRES IFS is using an octahedral reduced Gaussian grid, translating to a resolution from 8 km at the equator to 10 km at 70° N and 70° S before decreasing again towards the poles (Malardel et al., 2016). The output fields are available at a temporal resolution of 1 hour. Here we use two different configurations of IFS output to drive FLEXPART for times before 2021-01-01 and after. For the first period,

3-hourly IFS fields at 0.2° x 0.2° resolution for the Alpine area (4° to 16° E and 39° to 51° N; IFS-Alps) and 1° x 1° elsewhere were used, whereas afterwards hourly data at 0.1° x 0.1° resolution (-15° to 31° E to 36° to 61° N; IFS-EU) and 3-hourly global fields at 0.5° x 0.5° resolution were used.

### 2.5   FLEXPART LPDM

FLEXPART has been initially designed for estimating the mesoscale and synoptic dispersion of radio-nuclei from point

sources, such as releases during a nuclear accident like Chernobyl. Nowadays, FLEXPART (Stohl et al., 2005; Pisso et al., 2019), and other LPDMs (Jones et al., 2007), are utilized for a large variety of tracer transport problems, simulating the transport, diffusion, conversion, and deposition of various compounds ranging from inert GHGs to aerosol particles.

One of FLEXPART's major applications is in inverse modeling studies for the estimation of regional/continental-scale emissions of atmospheric compounds (Fang and Michalak, 2015; Henne et al., 2016; Brunner et al., 2012; Stohl et al., 2010).

This is due to FLEXPART's ability for both forward and backward in-time simulations. For backward simulations, particle trajectories are integrated backward in time, using a negative time step. The final product is an estimate of the sensitivity of a concentration measured at the receptor $y_i$ to an emission source $x_i$, called the source-receptor relationship (Seibert and Frank, 2004). Source-receptor relationships derived from FLEXPART are linear since all atmospheric processes considered during the transport of the tracers are linear (advection, diffusion, convective mixing). The compounds we are interested

in possess very long atmospheric lifetimes (5 years or more), so for the regional scale transport (less than 10 days) we can assume these to be inert. Thus, linear relationships, $m_{i,l}$, in units of $\mathrm{s\,m^3\,kg^{-1}\,mol\,mol^{-1}}$ with $i$ referring to different grid cells, and $l$ referring to different receptors, can be easily derived from FLEXPART. If the spatial distribution of emissions $E_i$ is multiplied by source sensitivities, $m_{i,l}$, the product yields the mixing ratio increment, $y_l$, of the tracer at the receptor site, $l$, resulting from emissions in the considered domain and time window

$$y_l = \sum_i m_{i,l} E_i,$$    (2)

to which the baseline concentration $y_{b,l}$ needs to be added to obtain the absolute mixing ratio. In our case, $y_{b,l}$ was estimated from observations using the REBS method (Sect. 2.3).





Here, we utilize two versions of FLEXPART in backward mode; FLEXPART-COSMO and FLEXPART-IFS. FLEXPART-COSMO is a version of FLEXPART adapted to the COSMO model (Henne et al., 2016). The meteorological fields driving FLEXPART
are directly used in the hybrid-z coordinate system of COSMO with no additional interpolation. FLEXPART-IFS interpolates the meteorological fields from the hybrid pressure coordinate system of ECMWF-IFS to a terrain-following z-based system (Stohl et al., 2005). The meteorological fields employed in FLEXPART simulations are some of the driving NWP's prognostic variables (winds, temperature, pressure, etc.) and accumulated fluxes (precipitation, surface heat, momentum, moisture).

FLEXPART-COSMO is employed at two different spatial resolutions, 7 km and 1 km (Sect. 2.4). When we refer to spatial
resolution, we always mean the resolution of the driving NWP, here COSMO, and not the LPDM. In the Lagrangian framework, there is no discretization of space and the frame of reference is centered on each particle following its trajectory in the space-time continuum.

Receptor-oriented FLEXPART simulations were carried out by releasing 50'000 particles at each different receptor continuously over 3-hour periods. Particles were then traced back for 8 days for FLEXPART-COSMO-7 and for 4 and 8 days for
FLEXPART-COSMO-1 coupled (see below) to FLEXPART-IFS, respectively. Source sensitivities were stored on two different output domains for FLEXPART-COSMO simulations: a larger domain (main, $0.16° \times 0.12°$ horizontal resolution) covering a similar area as the COSMO-7 simulations and a smaller and finer domain (nest, $0.02° \times 0.015°$ horizontal resolution) focusing on Switzerland. For simulations with FLEXPART-IFS, the output grid was at a lower horizontal resolution grid for the whole of Europe ($0.1° \times 0.1°$).

If FLEXPART is driven only by COSMO-1 fields, source sensitivities can only be produced for the limited COSMO-1 domain, and any European contributions from larger distances (as from the COSMO-7 domain) would be neglected. To account for this limitation, we offline nest FLEXPART-COSMO-1 to FLEXPART-IFS in order to continue the integration of the particles in Europe, once they leave the COSMO-1 domain (Katharopoulos et al., 2022).

Particle transport in FLEXPART is modelled by a simple zero acceleration scheme,

$$X(t+dt) = X(t) + u(X,t)dt, \tag{3}$$

$$u(X,t) = u_g(X,t) + u'(X,t) + u_m(X,t), \tag{4}$$

where $\mathbf{X}$ is the particle's position, $\mathbf{u}$ is the wind vector at the particle's location comprised of three components. The term $\mathbf{u}_g$ is the average wind vector at the particle location (in our case taken from the COSMO model), $\mathbf{u}'$ is the fluctuation from the mean wind representing the turbulence in the atmosphere (modeled as a stochastic Markov chain process), and $\mathbf{u_m}$
represents additional mesoscale wind variations (Stohl et al., 2005).

For simulations with FLEXPART-COSMO-7, we use the original turbulence parameterization of FLEXPART, the Hanna turbulence scheme (Stohl et al., 2005), while for FLEXPART-COSMO-1 we utilize the novel scheme introduced by Katharopoulos et al. (2022). We recently showed that since the Hanna scheme is developed to parameterize the whole turbulence spectrum, and COSMO-1 wind fields explicitly resolve part of the turbulence spectrum –eddies of the size of the model grid and



bigger can be represented by the wind fields– that leads to duplication of parts of the turbulence spectrum in the model, and as a result, to increased diffusion.

## 2.6  Inversion framework

As we have already mentioned, FLEXPART was utilized in the backward mode to produce source sensitivities, $\boldsymbol{M}$, which translate spatial emissions, $\boldsymbol{x}$, to mole fractions, $\boldsymbol{y}$, at the receptor site

$$\boldsymbol{y} = \boldsymbol{M}\boldsymbol{x}. \tag{5}$$

The state vector $\boldsymbol{x}$, $\boldsymbol{x} = (x_1, ..., x_k)^T$, contained K elements, which correspond to the sum of the total number of grid cells, $N_E$, in our inversion grid and the total number of baseline nodes, $N_B$, to be optimized by the inversion. The baseline nodes are baseline factors at discrete time intervals, since we do not optimize the baseline at every time step to reduce the size of the matrix, $\boldsymbol{M}$, and also avoid ending up with an under-determined system of equations. Here, the time interval between

our baseline nodes was set to $\tau_b = 30$ days. The estimation of the a-priori baseline is described in Sect. 2.3. The rectangular matrix $\boldsymbol{M}$ (size $L \times K$) is a column block matrix with two blocks, $\boldsymbol{M_E}$ and $\boldsymbol{M_B}$, representing the sensitivity of the observations to emissions for each grid cell and the baseline mole fractions, respectively. The mole fractions at the receptor sites are the product of the sensitivity matrix with the state vector, and its length is equal to the number of observations at all receptor sites, $\boldsymbol{y} = (y_1, ..., y_L)^T$.

If we would use the complete output grid of our transport model as the inversion grid, then the size of our sensitivity matrix would be too large to be computationally manageable and the solution probably would be under-determined depending on the spatial correlation lengths. Fine grids with negligible source sensitivities and very low a-priori emissions are also more prone to be assigned negative emissions in typical dipole patterns since we assume Gaussian distributed errors. To reduce the size of the inversion problem, an irregularly sized inversion grid is introduced that assigns finer (lower)

grid cells in areas with larger (smaller) average source sensitivities (Henne et al., 2016). The number of grid cells in our inversions varies from 1000–2500 depending on the number of observations available for different inversions.

Bayesian inverse modeling is employed to statistically optimize the estimates of the variables of interest, $\boldsymbol{x}$, by constraining them with the observational data, $\boldsymbol{y^0}$ (top-down constraint), and with the prior estimate of the variables of interest, $\boldsymbol{x_b}$ (bottom-up constraint). Gaussian distributed errors are always assumed between the observations and the simulated mole

fractions and between the a-priori and the a-posteriori emissions,

$$P(\boldsymbol{x}) = \frac{1}{\sqrt{2\pi |\boldsymbol{B}|}} e^{-\frac{1}{2}(\boldsymbol{x}-\boldsymbol{x_b})^T \boldsymbol{B}^{-1}(\boldsymbol{x}-\boldsymbol{x_b})}, \tag{6}$$

$$P(\boldsymbol{y^0}|\boldsymbol{x}) = \frac{1}{\sqrt{2\pi |\boldsymbol{R}|}} e^{-\frac{1}{2}(\boldsymbol{y^0}-\boldsymbol{M}\boldsymbol{x})^T \boldsymbol{R}^{-1}(\boldsymbol{y^0}-\boldsymbol{M}\boldsymbol{x})}, \tag{7}$$

where $\boldsymbol{B}$ is the a-priori error covariance matrix, and $\boldsymbol{R}$ is the observational error covariance matrix. The construction of these matrices is discussed in Sect. 2.6.1. By applying Bayes theorem, $P(\boldsymbol{x}|\boldsymbol{y}) \approx P(\boldsymbol{y}|\boldsymbol{x})P(\boldsymbol{x})$, we obtain the a-posteriori





Gaussian probability distribution function for the error of the emissions. The cost function, Equation 8, is the negative logarithm of $P(\boldsymbol{x}|\boldsymbol{y})$. We minimize the cost function to find the value of the state of the emissions, $x$, that minimizes the observational and a-priori error

$$J(\boldsymbol{x}) = (\boldsymbol{y^0} - \boldsymbol{Mx})^T \boldsymbol{R}^{-1} (\boldsymbol{y^0} - \boldsymbol{Mx}) + (\boldsymbol{x} - \boldsymbol{x_b})^T \boldsymbol{B}^{-1} (\boldsymbol{x} - \boldsymbol{x_b}). \tag{8}$$

The minimization problem can be solved analytically, since the sensitivity matrix, $\boldsymbol{Mx}$, is a linear mapping, $\boldsymbol{x} \in \mathbb{R}^K \rightarrow$
$\boldsymbol{Mx} \in \mathbb{R}^L$. Major advantages of the analytical approach are 1) the complete characterization of the a-posteriori error (Brasseur and Jacob, 2017) as part of the solution, and 2) that it can be fast and well suited for a plethora of sensitivity inversions. The minimization of the cost function yields the solution,

$$\hat{\boldsymbol{x}} = \boldsymbol{x_b} + \boldsymbol{G}(\boldsymbol{y} - \boldsymbol{Mx_b}), \tag{9}$$

where $\boldsymbol{G}$ is the gain matrix,

$$\boldsymbol{G} = \boldsymbol{B}\boldsymbol{M}^T (\boldsymbol{M}\boldsymbol{B}\boldsymbol{M}^T + \boldsymbol{R})^{-1}, \tag{10}$$

giving the sensitivity of the optimal state to the observations. In the analytical inversion, the a-posterior error covariance matrix can be directly calculated as,

$$\hat{\boldsymbol{S}} = \boldsymbol{B} - \boldsymbol{B}\boldsymbol{M}\boldsymbol{G}, \tag{11}$$

describing the uncertainty of the posterior estimate.

**2.6.1  Covariance matrices**

Our design of the error covariance matrices, $\boldsymbol{B}$ and $\boldsymbol{R}$, follows a maximum likelihood approach for which initial estimates of the matrices are needed (Henne et al., 2016). Both covariance matrices are symmetric block matrices. The observational error covariance matrix, $\boldsymbol{R} = [\boldsymbol{\epsilon_o}\boldsymbol{\epsilon_o^T}]$, contains contributions from the instrument error, $\boldsymbol{\epsilon_I}$, the representation error, $\boldsymbol{\epsilon_R}$, and the model error, $\boldsymbol{\epsilon_M}$. These errors are assumed to be uncorrelated, so the covariance matrix can be calculated as the sum
of squares of the individual covariance matrices for each source of error, $\boldsymbol{\epsilon_o}^2 = \boldsymbol{\epsilon_I}^2 + \boldsymbol{\epsilon_R}^2 + \boldsymbol{\epsilon_M}^2$.

The block matrix $\boldsymbol{R}$ is a row block matrix, containing a number of blocks equal to the number of different receptors. Diagonal elements of $\boldsymbol{R}$ are estimated as follows:

$$R_{i,i} = \epsilon_I^2 + \alpha + \beta\chi_{p,i}^2. \tag{12}$$

Representation and model errors are considered as a single error, increasing linearly with a-priori-simulated mixing ratios,
$\chi_{pi}$. The factors $\alpha$ and $\beta$ are determined by the log-likelihood approach. For each block matrix representing an individual receptor, temporal correlation in the error is added to the covariance matrix by setting the non-diagonal entries to

$$R_{i,j} = e^{-\frac{T_{i,j}}{\tau_0}} \sqrt{R_{i,i}} \sqrt{R_{j,j}} \quad i \neq j. \tag{13}$$



The factor $T_{i,j}$ is the time difference between measurements, and $\tau_0$ is the temporal correlation length, here set to a very small value of 0.01 days, meaning that there is very low auto-correlation between the daily average observations as used in the inversion. Error correlation between different sites is neglected.

The matrix $\boldsymbol{B}$ consists of two block matrices. The first corresponds to the emissions, $\boldsymbol{B}^E$, and the second to the baseline, $\boldsymbol{B}^B$. The diagonal elements of matrix $\boldsymbol{B}^E$ are proportional (factor $f_E$) to the a-priori emissions, while the off-diagonal elements are spatially correlated. The correlation fades as an exponential function of their distance, $d_{i,j}$, scaled by a correlation length scale, $L$,

$$B^E_{i,i} = \left(f_E x_{b,i}\right)^2 \tag{14}$$

$$B^E_{i,j} = e^{-\frac{d_{i,j}}{L}} \sqrt{B^E_{i,i}} \sqrt{B^E_{j,j}} \quad i \neq j. \tag{15}$$

The diagonal values of block matrix $\boldsymbol{B}^B$ are proportional (factor $f_b$) to the baseline error, while the non-diagonal elements are set to be correlated in time. The correlation fades as an exponential function of the time difference between baseline nodes, $T_{i,j}$, scaled by a temporal correlation length, $\tau_b$

$$B^B_{i,i} = \left(f_b \sigma_b\right)^2 \tag{16}$$

$$B^B_{i,i} = e^{-\frac{T_{i,j}}{\tau_b}} \sqrt{B^B_{i,i}} \sqrt{B^B_{j,j}} \quad i \neq j. \tag{17}$$

### 2.6.2 Maximum likelihood

Accurate knowledge of the a-priori and observational error covariance matrices, $\boldsymbol{B}$ and $\boldsymbol{R}$, is, in general, unavailable and often 'expert judgments' are used to estimate or set the parameters describing the matrices. Similarly, our initial values of the covariance matrices are a mix of expert judgment and methods used in the literature (Henne et al., 2016). To overcome the partial subjectivity of the construction of the covariance matrices, we employ a maximum likelihood optimization step (Michalak et al., 2005). The parameters that we optimize in the maximum likelihood optimization are the correlation length, $L$, the factor $f_E$, which gives the variance of the emissions at each grid cell relative to the prior emissions, and the temporal correlation length of the baseline, $\tau_B$. Additionally, for each different receptor, the factors $f_b$, $\alpha$, and $\beta$ are optimized. The maximum likelihood estimate of the covariance parameters is obtained by minimizing Eq. 18 with respect to the covariance parameters (Michalak et al., 2005),

$$L_\theta = \frac{1}{2} ln|\boldsymbol{M}\boldsymbol{B}\boldsymbol{M}^T + \boldsymbol{R}| + \frac{1}{2}(\boldsymbol{y} - \boldsymbol{H}\boldsymbol{x_b})^T (\boldsymbol{M}\boldsymbol{B}\boldsymbol{M}^T + \boldsymbol{R})^{-1}(\boldsymbol{y} - \boldsymbol{M}\boldsymbol{x_b}). \tag{18}$$

### 2.7 Sensitivity tests

The main focus of this study is to assess the impact of high-resolution FLEXPART-COSMO-1 simulations on the emission estimates of halocarbons. The transport model resolution is one of the factors which can influence the total inverse emission estimates, their spatial distribution, and their uncertainty. The kind of analytical inversion used here to optimize the





emissions was shown to likely underestimate the uncertainty of the a-posteriori state vector (e.g., Berchet et al., 2015). To
capture the whole range of uncertainty of our a-posteriori, we conducted additional sensitivity tests (Table 2), in which
we vary different parameters and aspects of our inversion (transport model, inversion grid, spatial distribution of a-priori
emissions, optimization of different covariance parameters during the maximum likelihood estimation step, temporal res-
olution of the assimilated observations, sensitivity of the inversion to the inclusion of observations from additional sites,
seasonality of emissions). In the following, our BASE inversion (Table 2) corresponds to inversions for which BRM, SOT,
and JFJ are employed as the observational sites in Switzerland. TAC and MHD are used in this setup as additional non-
Swiss observational sites. Furthermore, the maximum likelihood step is calculated for all covariance parameters except $L$
and $f_b$, which are fixed to specific values for each compound. The observations are aggregated over 24-hour intervals and
the irregular grid size is increased to approximately 2000 grid cells, compared to the inversions with fewer cells in Rust et al.
(2022). This setup is used for comparisons across inversions with different transport model resolutions for the same tracer
(BASE1 and BASE7, Table 2). All the different sensitivity tests described in the following sections are summarized in Table
2.

### 2.7.1 Transport model

Two versions of FLEXPART are used in this study, FLEXPART-COSMO-7, and FLEXPART-COSMO-1. Fig. 2 depicts the foot-
prints or source sensitivities for the two different setups and for the two different observational sites used on the Swiss
Plateau, BRM, and SOT. The footprints of the two models exhibit similar distributions on the Swiss Plateau, but they dif-
fer significantly in the Alpine region. The higher resolution model, FLEXPART-COSMO-1, is able to depict the flow in the
Alpine valleys because of the better representation of the topography. On the Swiss Plateau, both models present their high-
est sensitivities close to the receptors, and their sensitivities decay close to the Swiss borders. The highest values of SOT
footprints for the high-resolution model are focused on the region around SOT, while the low-resolution model extends
the high sensitivities towards the canton of Valais, Geneva, and the middle of the Swiss Plateau.

### 2.7.2 Spatial distribution of a-priori emissions

We conducted inversions using three different spatial distributions of the a-priori emissions, $x_b$, in order to test the sen-
sitivity of the a-posteriori estimated vector and its uncertainty to different a-priori choices (Table 2). Please note that in-
dependent of the spatial distribution of the emissions, the probability distribution of each element of the state vector $x_b$
always follows a Gaussian distribution in all our analytical inversions. The a-priori emissions for individual countries were
taken from the annual national inventory reports (NIR) to the United Nations Framework Convention on Climate Change
(UNFCCC) for the reference year 2018 as reported in April 2020.

For some widely-used substances, such as HFC_134a, we expect that the usage and, hence, the emissions, mostly follow
proxies like population and traffic (HFC-134a in mobile air conditioning). For other compounds, such as $SF_6$, used as
insulator gas in high-voltage installations, the choice of the a-priori is not as obvious since their emissions may be more
dominated by individual emission hot spots. For these substances, using different a-priori fields allows for illustrating how





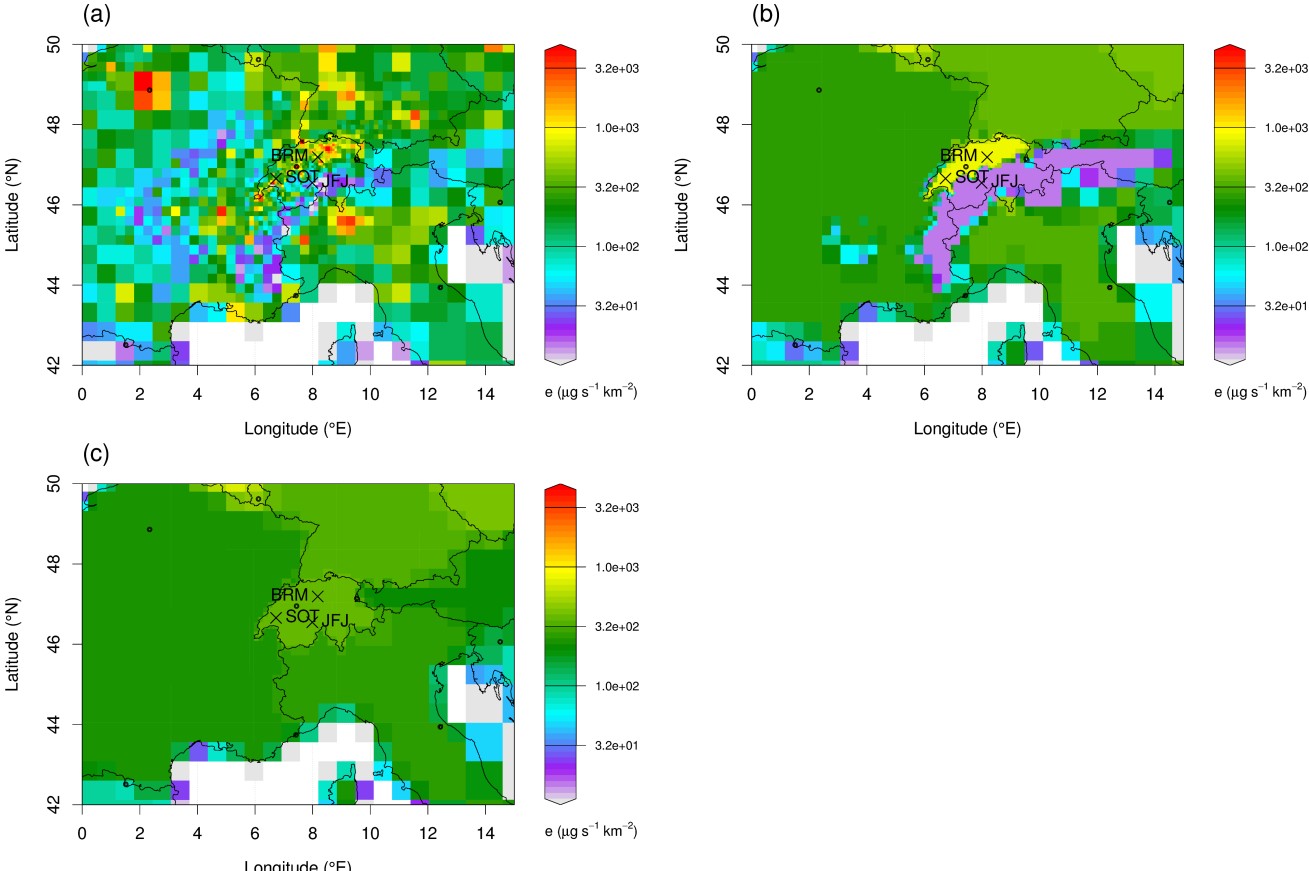

**Figure 3.** Different a-priori spatial emission distributions –presented in the irregular grid– utilized in the inversions. Population-based a-priori can be seen in (a), elevation-dependent a-priori in (b), and uniformly distributed emissions per country in (c).

strongly the inverse solution is guided by the a-priori, and reveals if the higher resolution transport model inhibits a larger potential to localize emissions independent of the a-priori.

The different a-priori fields used in this study consist of a population-based a-priori, a uniform-per-country a-priori, and

an elevation-dependent a-priori. In the population-based a-priori, an emission factor represents the average emissions for each person in the country, and the emissions are given by the emission factor multiplied by the number of residents in each grid cell. In the uniform-per-country case, the emissions are distributed uniformly in the whole country, while in the elevation-dependent a-priori, the emissions are distributed uniformly per country below an elevation threshold of 1000 m, whereas above that threshold the emissions were set to 5 % of the low elevation value. Above the elevation threshold,

population densities are usually low in the Alps and very few industrial installations are present, suggesting very limited emissions of the current substances of interest. The spatial distribution of the different a-priori emissions can be seen in Fig. 3.





### 2.7.3 Covariance parameters and baseline uncertainty

As already mentioned, the parameters optimized in the maximum likelihood optimization are the correlation length, $L$,
the factor $f_E$, which scales the variance of the emissions at each grid cell, the temporal correlation length, $\tau_B$, and for
each different receptor, the factors $f_b$, $\sigma_M$, and $\sigma_R$. The factor $f_b$ scales the uncertainty of the baseline. The maximum
likelihood method (Sect. 2.6.2) was employed for all the parameters except the correlation length, $L$, and the uncertainty
scaling factor, $f_b$, since they significantly alter the emission estimates. The latter two parameters were set to fixed values
for each different compound, so the low- and high-resolution inversions are comparable (Table 2).

### 2.7.4 Inversion grid

As already mentioned in Sect. 2.5, the output grid size of the inversion varies with respect to source sensitivities. In re-
gions with low sensitivity, FLEXPART's output grid cells are aggregated to form bigger grid cells. We conducted sensitivity
inversions to assess whether different grids with a varied number of cells result in different spatial distributions and to-
tal emissions in Switzerland (Table 2). This is of special importance since two of the anticipated emission hot spots in
Switzerland (cities of Zurich and Lausanne) are not very distant from the observational sites at BRM and SOT, respectively.

### 2.7.5 Observational sites

The sensitivity of total Swiss emissions and their spatial distribution to additional observation sites inside and outside
Switzerland was further explored. Long-term halocarbon observations are only available from the AGAGE network. We
further employed data for Switzerland from the two field campaigns in Beromünster (2019-2020), and in Sottens (2021).
The sensitivity of the emissions to the inclusion of observations from Beromünster or Sottens, or from both sites, was
further explored. In our BASE inversions, the non-Swiss receptors used are TAC in UK and MHD in Ireland. Inversions with
additional observations from TOB and CMN were conducted for HFC-134a to test the sensitivity of Swiss emissions to the
inclusion of additional sites closer to Switzerland (Table 2).

### 2.7.6 Seasonal variability

In our BASE inversions, the total emissions and their spatial distribution represent average values over the whole year;
no annual cycle is considered. For refrigerants such as HFC-134a and HFC-125, this assumption can be ambiguous. HFC-
134a is mainly used in mobile air conditioning in cars, but we do not know if the emissions are stronger when the air
conditioning system is in use (mainly in summer months), or if they are at a constant rate independent of the usage. There
is some evidence in the literature supporting a seasonal cycle of the emissions (Xiang et al., 2014; Hu et al., 2015). To test
the impact of this assumption on the total emissions and whether seasonality is revealed by the inversion system, we
conducted a sensitivity inversion for HFC-134a extending the emissions state vector to separately hold emissions for each
different season. The seasons were defined according to the meteorological definition. In our sensitivity inversion, the a-



**Table 2.** Different groups of inversions conducted in this study.

| Inversion ID | Sensitivity variation | Receptors | Transport model | HFC-134a | HFC-125 | HFC-32 | SF$_6$ |
|---|---|---|---|---|---|---|---|
| **Base inversions** | | | | | | | |
| BASE7 | BASE | ( BRM, SOT, JFJ, | C7 | x | x | x | x |
| BASE1 | BASE | MHD, TAC) | C1 | x | x | x | x |
| **Sensitivity inversions** | | | | | | | |
| BASE_ED7 | A-priori distribution | ( BRM+SOT+JFJ | C7 | x | x | x | x |
| BASE_ED1 | (elevation-dependent) | +MHD+TAC) | C1 | x | x | x | x |
| BASE_UNI7 | A-priori distribution | ( BRM, SOT, JFJ, | C7 | x | x | x | x |
| BASE_UNI1 | (uniform) | MHD, TAC) | C1 | x | x | x | x |
| SEAS1 | Emission variability | ( BRM, SOT, JFJ, | C1 | x | - | - | - |
| SEAS2 | (seasonal) | MHD, TAC) | C1 | x | - | - | - |
| **Preliminary screening** | | | | | | | |
| PREL_COV7 | Covariance parameters | ( BRM, SOT, JFJ, | C7 | x | x | x | x |
| PREL_COV1 | (optimize $L$ & $f_b$) | MHD, TAC) | C1 | x | x | x | x |
| PREL_NCEL7 | Inversion grid | ( BRM, SOT, JFJ, | C7 | x | x | x | x |
| PREL_NCEL1 | (number of cells) | MHD, TAC) | C1 | x | x | x | x |
| PREL_SITEXT7 | Observational sites | ( BRM, JFJ, MHD, | C7 | x | - | - | - |
| PREL_SITEXT1 | (incl. CMN & TOB) | TAC, CMN, TOB) | C1 | x | - | - | - |
| PREL_SITRED7 | Observational sites | ( BRM, JFJ, | C7 | x | x | x | x |
| PREL_SITRED1 | (excl. SOT) | MHD, TAC) | C1 | x | x | x | x |
| PREL_AGR7 | Observation aggregation | ( BRM, SOT, JFJ, | C7 | x | - | - | - |
| PREL_AGR1 | (3-hourly) | MHD, TAC) | C1 | x | - | - | - |

priori emissions and their uncertainty were constant during the different seasons. Furthermore, we assumed a temporal correlation length scale for the a-priori covariance of 30 days (see Eq. 15).

Since running the maximum likelihood optimization for the enlarged inversion problem proved to be computationally too costly, two sensitivity inversions with slightly different covariance settings were performed: one with the covariance parameters taken directly from the outputs of the BASE inversion with maximum likelihood optimization (SEAS1) and one with the model error being determined by an iterative approach (SEAS2), as described in Stohl et al. (2010); Henne et al. (2016).



### 2.7.7 Observation aggregation

Finally, the sensitivity of the inversion to the temporal aggregation window of the assimilated observations was assessed. In our BASE inversions, we use observations averaged over 24-hour intervals. Since the high-resolution model was shown to improve the simulated representation of the observed diurnal cycle of tracer mole fractions at the BRM tall tower (Katharopoulos et al., 2022), we further performed sensitivity inversions employing 3-hourly aggregated observations ( Table 2) to investigate whether we obtain additional information from the sub-daily observed tracer variability.

## 3 Results

### 3.1 Preliminary screening tests

Swiss halocarbon emissions using the low-resolution model were estimated after the measurement campaign in Beromün-ster in 2020 and the results are summarized in Rust et al. (2022). The inversions conducted for their study included obser-vations from two sites in Switzerland, JFJ, and BRM, and two sites on the British Isles, MHD and TAC, to constrain the European emissions. Here, we first examine the impact of the additional observational sites (CMN and TOB) on the Swiss national emission estimates. Sensitivity inversions were conducted for both the high- and the low-resolution models and for HFC-134a including observations from CMN and TOB (PREL_SITEXT7 and PREL_SITEXT1, Table 2). Only results from the high-resolution model are discussed in the following. In Fig. 4, the spatial distributions of the a-posteriori emissions of HFC-134a are displayed for the inversion excluding (PREL_SITRED1), (a), and including CMN and TOB (PREL_SITEXT1), (b), and the resulting difference (c). For HFC-134a no significant spatial differences between the a-posteriori emissions of the two inversions can be seen (Fig. 4, c). The differences in the total Swiss emissions for the two inversions are also not significant, $308\pm48\,\mathrm{Mg\,yr^{-1}}$ for the BASE inversion and $312\pm50\,\mathrm{Mg\,yr^{-1}}$ when including CMN and TOB.

The same cannot be claimed for observational sites in Switzerland though, since the PREL_SITRED1 inversion changes significantly both in terms of spatial distribution and total emissions when SOT (BASE1) is included (Sect. 3.2, Fig. 4 panel d). This is an expected result since SOT adds sensitivity to regions where BRM is not very sensitive (Fig. 2). Additional observational sites are also essential, since they allow for sampling emissions from the same region at different sites and hence under different atmospheric conditions (advection diretion, turbulence regime), thereby improving the represen-tation of dispersion. This happens because turbulent dispersion behaves differently in the near- and the far-field. In the near-field, dispersion approaches isotropy both at the large and small scales, meaning that the diffusion in the near-field is independent of the size of the eddies.

Concerning the covariance parameters which were excluded from the maximum likelihood step (Sect. 2.7.3), $f_b$ was ini-tially set to 1, meaning that the baseline is assigned an uncertainty equal to the uncertainty calculated in the REBS method (Sect. 2.3). The latter sometimes leads to unrealistically large adjustments in the baseline, and usually underestimation of the emissions, since most of these adjustments tend to increase the baseline considerably. Different sensitivity tests with different values of the factor $f_b$ were conducted to find a representative value for each different receptor and for each





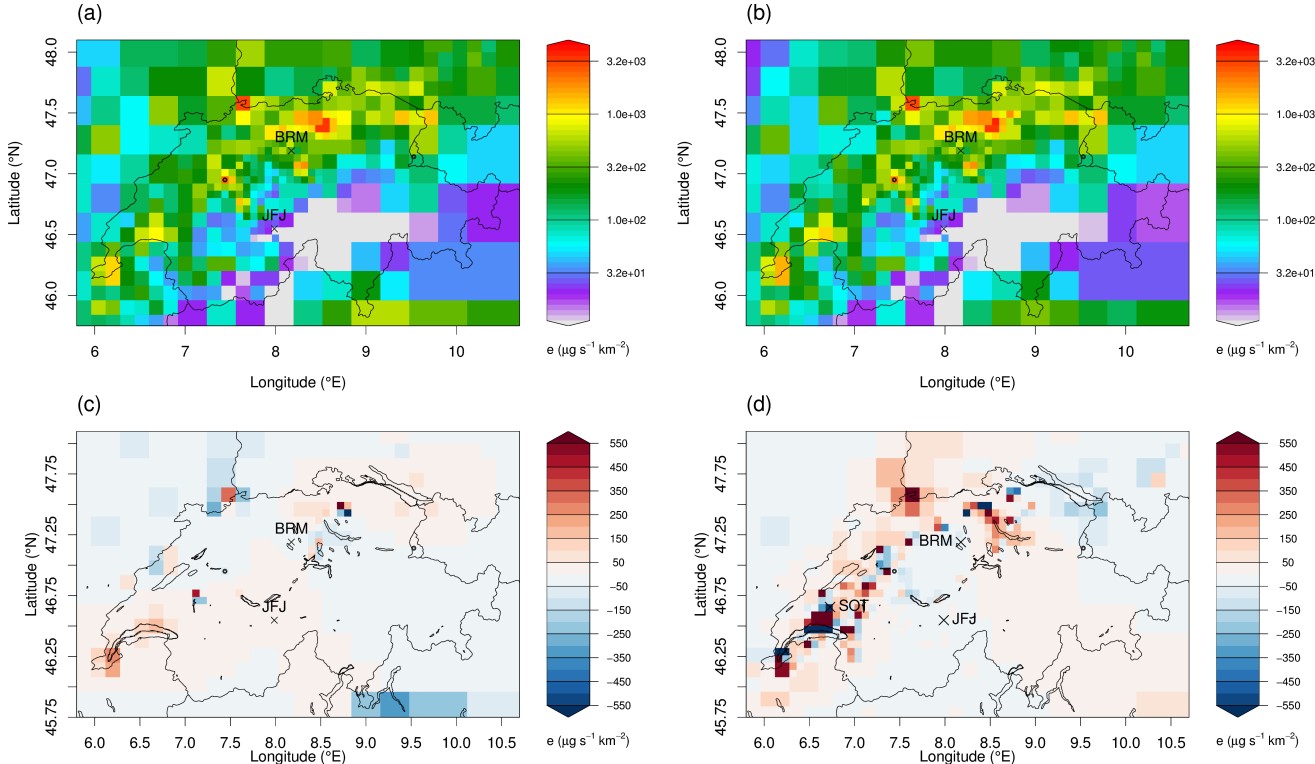

**Figure 4.** Spatial distribution of Swiss HFC-134a a-posteriori emissions for the inversion PREL_SITRED1 (a), and for PREL_SITEXT1 (b). (c) shows the a-posteriori emission differences between (a) and (b), while in (d) the a-posteriori emission differences between (a) and BASE1 inversion is shown. In all cases, results from the high-resolution transport model are given.

different compound (PREL_COV1 and PREL_COV7). Then, for all the inversions for this compound, the value of the factor $f_b$ was fixed, and not further optimized in the maximum likelihood step. Another factor that is poorly constrained by the maximum likelihood approach is the correlation length, $L$. Values pointing to overfitting, were also obtained from the

495 maximum likelihood, mainly for the low-resolution model. A series of sensitivity runs were deployed for the estimation of a meaningful correlation length, which afterward was used as a fixed value in the inversion for both model resolutions without being further optimized by the maximum likelihood (PREL_COV1 and PREL_COV7). The total emission estimates were not sensitive to small to medium deviations of the correlation length from the chosen value.

Furthermore, we investigated whether we obtain additional information from the high-resolution inversions if we use

3-hourly observation aggregates to drive the inversion instead of 24-hourly aggregates, as used for the low-resolution inversions and in previous studies (Rust et al., 2022) (PREL_AGR1 and PREL_AGR7). There was no significant difference when the 3-hourly aggregates were used, so we maintained the 24-hourly aggregates for the inversions in this study since they result in considerably reduced computational costs.



Finally, we fixed the parameters, which influence the resolution of the inversion grid, to values that lead to similar in-
version grids for both FLEXPART model resolutions (PREL_NCEL1 and PREL_NCEL7). However, the total country or total
inversion emissions did not show sensitivity to the resolution of the inversion grid (<1% differences across the two models)
within the range of tested resolutions.

## 3.2 Emissions of HFC-134a

HFC-134a is the most used halocarbon/HFC in Switzerland with reported emissions of $455 \, \mathrm{Mg \, yr^{-1}}$ for 2019 and $415 \, \mathrm{Mg \, yr^{-1}}$
for 2020 (FOEN, 2022). HFC-134a is employed as a refrigerant both in mobile air conditioning (i.e., road traffic and in sta-
tionary refrigeration systems). It is also used as a foam blowing agent. Its 100-year GWP is 1430 and its atmospheric lifetime
is approximately 14 years (Engel et al., 2018).

BASE7 inversion leads to an a-posteriori estimate of annual Swiss emissions of $260\pm49 \, \mathrm{Mg \, yr^{-1}}$ (Fig. 9). The a-posteriori
distribution of the emissions for this inversion can be seen in panel (a) of Fig. 5 and the difference between the a-posteriori
values and the a-priori in panel (a) of Fig. 6. Compared to the UNFCCC reported bottom-up emissions, there is a significant
reduction in HFC-134a emissions almost everywhere in Switzerland except for the region south of SOT and in the canton
of Valais, where the emissions are increased compared to the a-priori. In Rust et al. (2022) the Swiss emission estimate
for HFC-134a –using observations from BRM, JFJ, MHD, TAC, a population-based a-priori, and inversions with the low-
resolution model (PREL_SITRED1)– was $274 \pm 67 \, \mathrm{Mg \, yr^{-1}}$ (two standard deviations) for 2019–2020

The same a-posteriori emissions distributions but obtained with the high-resolution transport model can be seen in
Fig. 5 (b) and the difference from the a-priori emissions in Fig. 6 (b). The total emissions estimate for BASE1 inversion is
$351\pm44 \, \mathrm{Mg \, yr^{-1}}$ (Fig. 9), which is 35 % higher compared to the BASE7 estimate. The BASE1 estimate is closer to the value
of the inventory, while the BASE7 gives a 40 % lower estimate than the inventory. The relative uncertainty in the estimate is
higher for the BASE7, 12.8 %, compared to the BASE1, 18.8 %. This is an indication of an improved use of the information
content of the observations by the high-resolution model due to the improved representation of the atmospheric flow.





**Figure 5.** Spatial distribution of Swiss HFC-134a a-posteriori emissions for the BASE inversion with the 7 km model (a) and the 1 km model (b) starting from a population-based a-priori, and the same plots, (c) and (d), starting from a spatially uniform a-priori, and from an elevation-dependent a-priori, (e) and (f).

If we consider the difference between the a-posteriori emissions of the high- and the low-resolution model inversions (BASE1 and BASE7), the BASE1 inversion enhances the emissions in all big cities of Switzerland, in the regions with the most industrial activity (canton of Aargau, west north-west of Zurich), and along the traffic network Fig. 8. The arc with increased emissions from Zurich to Bern, Fig. 8, could point to emissions from the main highway of Switzerland (A1) form-





ing the main west-east transport route and connecting two of the biggest cities of Switzerland. To evaluate the connection between HFC-134a emissions and traffic, we calculated the correlation between a-posteriori emissions and $CO_2$ traffic emissions, as taken from the spatially-resolved Swiss emission inventory (Heldstab et al., 2021), for the two different inversions. The a-posteriori emissions from the BASE1 inversion show a higher correlation, r=0.6, with the traffic $CO_2$ emissions compared to the BASE7 inversion emissions, r=0.35. Since we start from a population-based a-priori, the correlation be-

tween the a-posteriori emissions and the population was additionally estimated. A-posteriori emissions from the BASE1 inversion possess a correlation of r=0.96 with the population, while the emissions from the BASE7 inversion are slightly less correlated, r=0.8. Hence, the high-resolution model inversion stays closer to the a-priori distribution compared to the low-resolution model.




**Figure 6.** A-posteriori minus a-priori emission differences for HFC-134a for the BASE inversion with the 7 km model (a) and the 1 km model (b) starting from a population-based a-priori, and the same plots, (c) and (d), starting from an elevation-dependent a-priori.

Fig. 7 shows the mole fraction timeseries of daily-averaged HFC-134a for both low- and high-resolution BASE inversions and the observations at BRM, (a), and SOT, (b). Both inversions represent the observed variability closely, however, some distinct differences do exist. To assess the performance of our inversions, we used the following statistical measures: reduced $\chi^2$ index, which is a measure of the normalized variance between the observations and the simulated values, the degrees of freedom, which is a measure of the relative uncertainty reduction between the a-priori and the a-posteriori, the



correlation coefficient, r, and the root mean square error (RMSE) of the simulated versus the observed mole fractions (Table 3). From these statistics for HFC-134a, we conclude that both BASE1 and BASE7 inversions are reliable, but the BASE1 inversions show improved performance at the receptors in Switzerland.

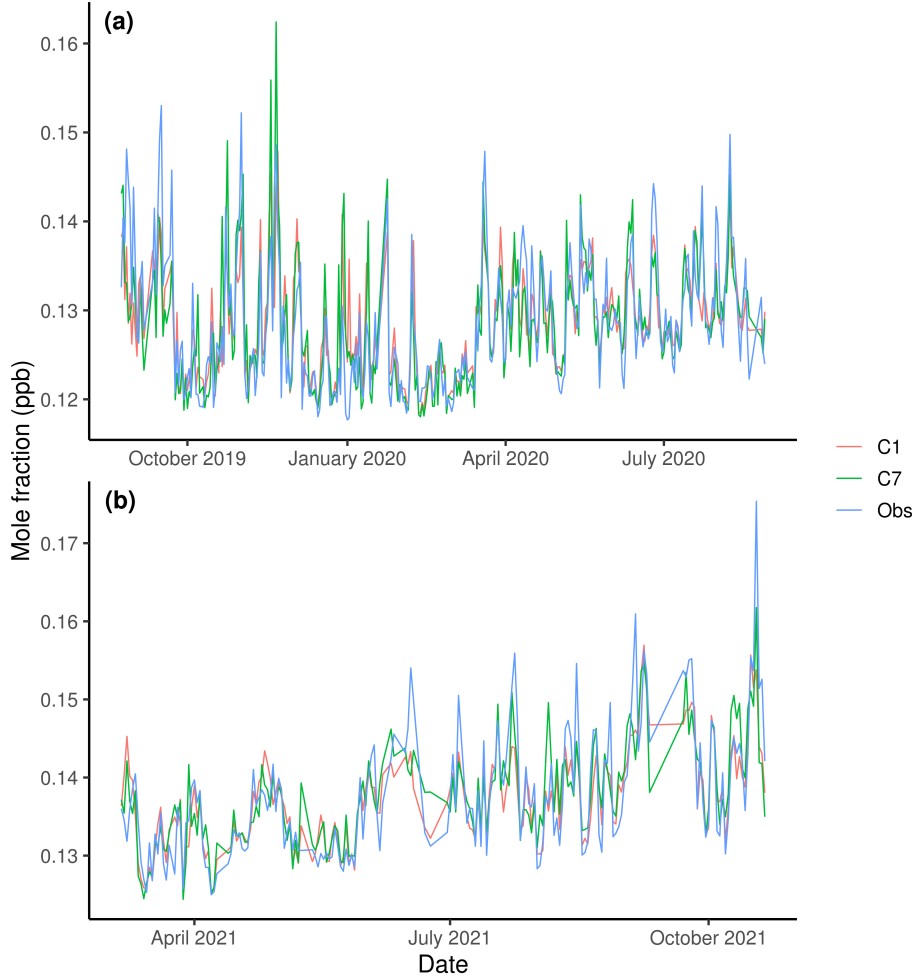

**Figure 7.** Time series of observed (blue lines) and simulated (red and green lines) HFC-134a mole fractions at BRM (a) and SOT (b). A-posteriori simulations for the BASE inversion and with the low- (C7, green lines) and high-resolution (C1, red lines) transport model are given.

For the inversions with uniformly distributed a-priori emissions by country (BASE_ED), (Figs. 5–6) panels (c) and (d), the results are inferior to the results obtained with the population-based a-priori. The BASE_ED inversions tend to retain and cannot completely remove the emissions from the Alpine region, while the distribution of emissions in the Swiss Plateau looks little plausible, at least for the low-resolution model. The inversion with the high-resolution model seems to be able to locate the emission hot-spots north of BRM and northwest of Zurich. These results highlight the importance of the a-priori





spatial distribution. If the inversion is initialized with a highly unrealistic a-priori and there is an insufficient observational constraint, the inversion may not converge to a rational state.

In Figs. 5 and 6, (e) and (f) correspond to inversions with an elevation-dependent a-priori (BASE_ED). The total Swiss
emission estimate is $318 \pm 62\,\mathrm{Mg\,yr^{-1}}$ for the BASE_ED1 and $217 \pm 46\,\mathrm{Mg\,yr^{-1}}$ for the BASE_ED7 (Fig. 9). For the inversions with the high-resolution model, Figs. 5 (f) and 6 (f), we can see that the inversion converges again towards a population-based distribution, especially close to the observational sites. The hot-spots of emissions in the cantons of Zurich and Aargau are reconstructed by the inversion, although not as sharply as for a population-based a-priori, along with the hot-spots in the Lausanne and Geneva regions. However, the inversion using the low-resolution transport model cannot recover
the population-based distribution to the same degree, Fig. 5 (c). Especially, the emissions from Zurich seem to be allocated too far to the west at a closer distance to BRM. Similarly, emissions from Geneva are not indicated as prominently. These observations are corroborated by the correlation between the a-posteriori emissions and population, which was r=0.56 for the high-resolution model and only r=0.31 for the low-resolution model. Since we are highly confident that the emissions of this substance should be correlated with population density, the BASE_ED inversions show that the high-resolution model
inversions are much more accurate in reconstructing the true distribution.

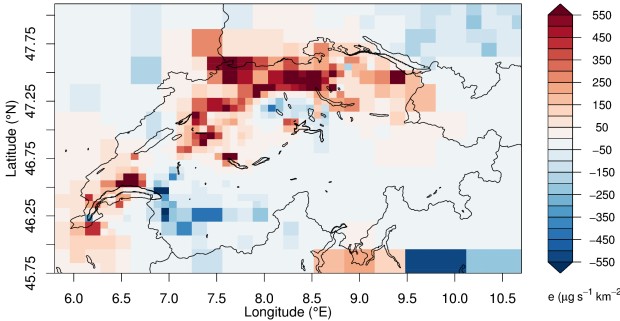

**Figure 8.** A-posteriori emission differences between the high- and low-resolution model inversions with population-based a-priori for HFC-134a.

Moreover, the HFC-134a inversions with seasonally variable emissions (SEAS) reveal the existence of a seasonality pattern in the emissions in Switzerland. In both types of seasonal inversions with the high-resolution model (2.7.6) there is a clear annual variability of HFC-134a with the peak during the summer months June–August (JJA)– $433 \pm 94\,\mathrm{Mg\,yr^{-1}}$ for SEAS1 and $364\,\mathrm{Mg\,yr^{-1}}$ for the SEAS2 inversion– and the minimum during the winter–$238 \pm 98\,\mathrm{Mg\,yr^{-1}}$ for the SEAS1
$239 \pm 100\,\mathrm{Mg\,yr^{-1}}$ for the SEAS2 inversion. This corresponds to a seasonal amplitude of approximately 1.3, which is similar to seasonal amplitudes obtained by Hu et al. (2017) for HFC emissions in North America. The spatial distribution of the emissions for the different seasons is similar, pointing to the conclusion that the emissions in all different seasons have the same sources, but the leakage of HFC-134a from refrigeration systems is higher when they are in use. According to the statistical measures used, the SEAS2 inversion is superior to the SEAS1, possessing a better correlation of the simulated



values when compared to the observations and reduced $\chi^2$ much closer to 1. The total annual emission estimate for the two inversions is $320 \pm 50 \, \mathrm{Mg\,yr^{-1}}$ for the SEAS2 $342 \pm 48 \, \mathrm{Mg\,yr^{-1}}$ for the SEAS1 inversion, close to the BASE estimate. Hence, there seems to be no big gain when we consider inversions with seasonality when the main target is the validation of annual total emissions. However, these simulations could help improve our understanding of the release mechanisms of these compounds.

For the low-resolution model, a much smaller (insignificant) seasonal amplitude was obtained in the a-posteriori emissions. Whether this is due to the reduced ability of the model to realistically reproduce the diurnal mole fraction variability as compared to the high-resolution model (Katharopoulos et al., 2022) or to potential seasonal transport biases will need to be investigated in future studies.

Based on the analysis in this section, we can claim that the high-resolution inversions reconstruct the spatial distri-
bution of the HFC-134a emissions in Switzerland better and with more detail than the low-resolution inversions. The total Swiss emission estimates between the two resolution models differ significantly, with the high-resolution model predicting values closer to those in the inventory.

For the remaining halocarbon inversions in this work, we present only the results from population-based a-priori and elevation-dependent a-priori since the elevation-dependent a-priori can be seen as an improved version of the uniform by
country a-priori. Figures for the simulations with a uniform spatial a-priori distribution can be found in the supplement.





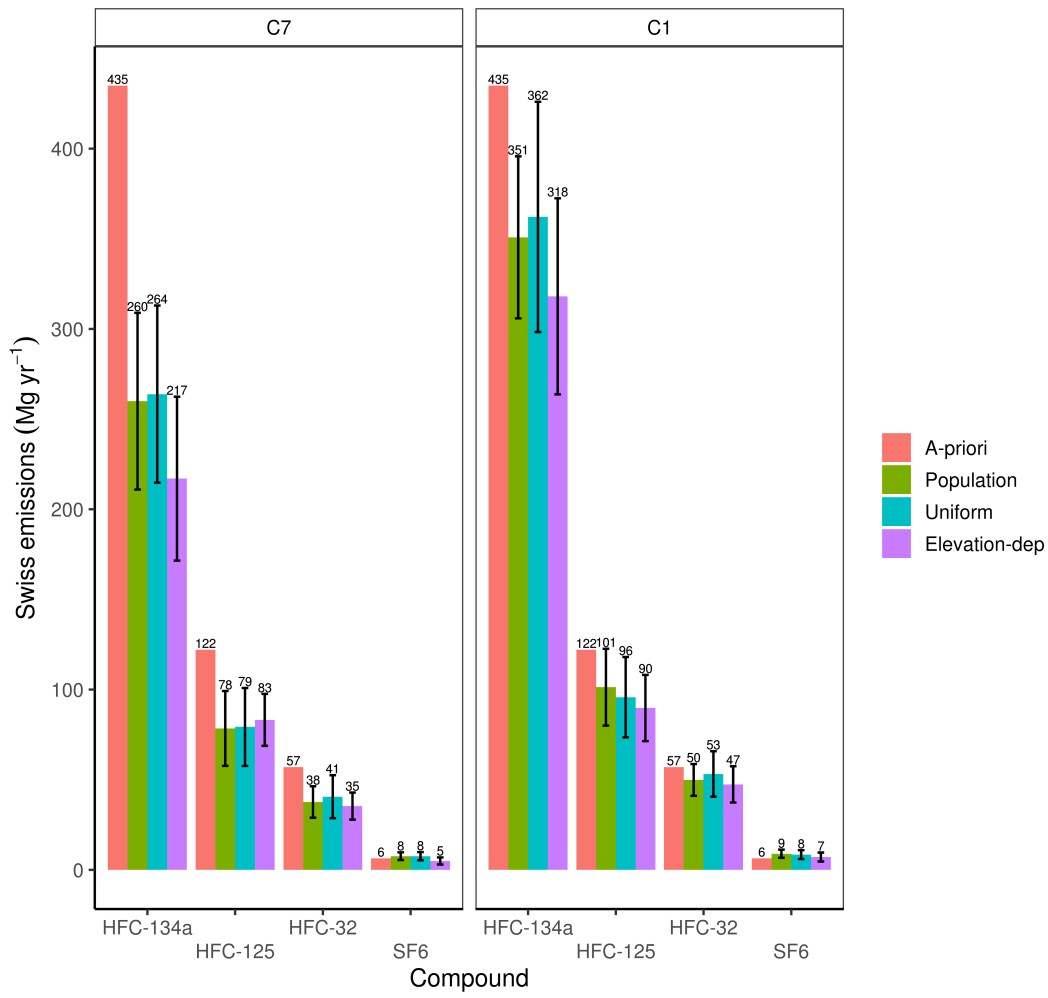

**Figure 9.** A-posteriori emissions for all the substances utilized in this study for the different model resolutions and the different a-prioris. All results correspond to the BASE inversions.

### 3.3 Emissions of HFC-125

HFC-125 is the second most abundant HFC in Switzerland, with reported emissions of 122 $Mg\,yr^{-1}$ for 2019 and 2020 (FOEN, 2022). HFC-125 is employed as a refrigerant mainly in stationary refrigeration systems, and as a result, its emissions are expected to be from static sources. It is also used as a fire suppression agent in fire extinguishers, but this use is forbidden in Switzerland. Although, on a mass basis, HFC-125 emissions are lower compared to HFC-134a, their impact as a GHG is higher since its 100-year GWP is 3500, almost three times that of HFC-134a. The setup used to estimate the Swiss emissions of HFC-125 is identical to the setup used for HFC-134a including the two Swiss sites (BRM, SOT).





**Table 3.** Statistical measures used to assess the reliability of inversion for different compounds, different transport model resolutions, and different a-priori emissions. The table displays the reduced $\chi^2$ index, degrees of freedom (DOF), root means squared error (RMSE), and correlations of simulated compound values against observations for BRM, SOT and JFJ.

| Compound | Model Res. | A-priori | $\chi^2$ | DOF | r (BRM) | r (SOT) | RMSE (BRM) (ppt) | RMSE (SOT) (ppt) |
|---|---|---|---|---|---|---|---|---|
| HFC-134a | C7 | Population | 1.01 | 88 | 0.74 | 0.75 | 4.66 | 4.63 |
| HFC-134a | C7 | Uniform | 1.08 | 88 | 0.71 | 0.65 | 4.91 | 5.70 |
| HFC-134a | C7 | Elevation-dep | 1.02 | 97 | 0.75 | 0.72 | 4.63 | 4.67 |
| HFC-134a | C1 | Population | 1.00 | 78 | 0.79 | 0.83 | 4.10 | 4.10 |
| HFC-134a | C1 | Uniform | 1.08 | 82 | 0.79 | 0.74 | 4.10 | 5.50 |
| HFC-134a | C1 | Elevation-dep | 1.01 | 91 | 0.8 | 0.83 | 4.00 | 4.10 |
| HFC-125 | C7 | Population | 1.02 | 98 | 0.76 | 0.77 | 1.18 | 1.17 |
| HFC-125 | C7 | Uniform | 1.07 | 97 | 0.73 | 0.72 | 1.26 | 1.26 |
| HFC-125 | C7 | Elevation-dep | 1.06 | 92 | 0.72 | 0.71 | 1.31 | 1.31 |
| HFC-125 | C1 | Population | 1.01 | 92 | 0.75 | 0.81 | 1.14 | 1.08 |
| HFC-125 | C1 | Uniform | 1.10 | 89 | 0.71 | 0.72 | 1.22 | 1.27 |
| HFC-125 | C1 | Elevation-dep | 1.06 | 93 | 0.72 | 0.78 | 1.20 | 1.17 |
| HFC-32 | C7 | Population | 1.02 | 94 | 0.68 | 0.75 | 1.57 | 1.71 |
| HFC-32 | C7 | Uniform | 1.09 | 86 | 0.67 | 0.70 | 1.59 | 1.87 |
| HFC-32 | C7 | Elevation-dep | 1.07 | 89 | 0.65 | 0.74 | 1.66 | 1.74 |
| HFC-32 | C1 | Population | 1.00 | 84 | 0.73 | 0.80 | 1.41 | 1.57 |
| HFC-32 | C1 | Uniform | 1.11 | 82 | 0.74 | 0.75 | 1.40 | 1.82 |
| HFC-32 | C1 | Elevation-dep | 1.04 | 90 | 0.74 | 0.81 | 1.39 | 1.55 |
| $SF_6$ | C7 | Population | 0.85 | 76 | 0.59 | 0.52 | 0.17 | 0.26 |
| $SF_6$ | C7 | Uniform | 0.85 | 54 | 0.49 | 0.42 | 0.18 | 0.28 |
| $SF_6$ | C7 | Elevation-dep | 0.86 | 73 | 0.62 | 0.52 | 0.15 | 0.27 |
| $SF_6$ | C1 | Population | 0.85 | 60 | 0.66 | 0.61 | 0.15 | 0.25 |
| $SF_6$ | C1 | Uniform | 0.87 | 60 | 0.67 | 0.56 | 0.15 | 0.26 |
| $SF_6$ | C1 | Elevation-dep | 0.85 | 72 | 0.69 | 0.60 | 0.14 | 0.25 |

The BASE7 inversion yields Swiss a-posteriori emissions of $78\pm20\,\mathrm{Mg\,yr}^{-1}$ (Figs. 10, 11), and (Fig. 9). In Rust et al. (2022) the Swiss emission estimate for HFC-125 with PREL_SITRED7 was $107\pm28\,\mathrm{Mg\,yr}^{-1}$ (two standard deviations) for 2019–2020 (Fig. S1–S2). The estimate from a second top-down method used in Rust et al. (2022) to estimate the emissions from BRM (tracer-ratio method) was $94\pm19\,\mathrm{Mg\,yr}^{-1}$. The addition of SOT yields approximately 20 % lower annual Swiss emis-





sions estimates. The a-posteriori minus a-priori emission difference figures for the two cases (not shown) depict that the PREL_SITRED7 inversion increases the emissions of HFC-125 compared to the a-priori north and north-west of BRM, in the Valais region, and in the regions north and south of SOT, whereas the BASE7 inversion including both BRM and SOT
increases the emissions compared to the a-priori only in a small radius around BRM and in the canton of Valais. In all other regions of the Swiss Plateau, a significant decrease in emissions is observed.

For the high-resolution BASE inversion, BASE1, the a-posteriori emissions can be seen in Fig. 10 (b) and the difference from the a-priori emissions in Fig. 11 (b). The total emissions estimate in this case is $101\pm21\,\mathrm{Mg\,yr^{-1}}$. This number is 22 % higher compared to the low-resolution model estimate. Similar to HFC-134a, the a-posteriori uncertainty is higher
(although only slightly) in the BASE7, 26 %, compared to the BASE1 inversion, 21 %. The BASE1 estimate is closer to the value of the inventory, whereas the inversion estimate with the BASE7 corresponds to about 2/3 of the inventory value. The BASE1 inversion increases the emissions in the region to the north of BRM, in Basel, and in eastern Switzerland close to the borders with Austria and Germany (Fig. 11 (b)). In contrast to the BASE7 inversion, the BASE1 inversion increases the emissions for all the big cities of Switzerland and in the industrial region ranging from Zurich to Basel (Fig. S3).

Additionally, in Figs. 10 and 11, (c) and (d), the a-posteriori emissions and the differences from the a-priori for HFC-125 can be seen starting from an elevation-dependent a-priori (BASE_ED). As with HFC-134a, the inversions converge again towards a population-based a-priori, that is especially close to the observational sites. The hot-spots of emissions in the cantons of Zurich and Aargau are reconstructed by the inversion, along with the hot-spots in the Lausanne and Geneva regions. The BASE_ED1 again tends to produce an a-posteriori distribution closer to the population distribution compared
to the BASE_ED7 inversion. The total Swiss emission estimate for the BASE_ED1 inversion with the elevation-dependent a-priori is $90\pm18\,\mathrm{Mg\,yr^{-1}}$, while that for the BASE_ED7 inversion $83\pm20\,\mathrm{Mg\,yr^{-1}}$. The results for the inversions starting from uniformly distributed emissions (BASE_UNI) can be seen in the supplement (Figs. S14–S17).

Both the high- and the low-resolution inversions are reliable (Table 3), since they present reasonable reduced $\chi^2$ and they both lower the uncertainty of the a-priori emissions (DOF). The high-resolution inversion for HFC-125 possesses a
slightly higher correlation and slightly smaller RMSE of simulated versus observed values at the Swiss receptors (Table 3). Since HFC-125 is used in stationary air conditioning systems, its usage should be concentrated in the big cities and in industrial areas and should partially follow a population-based distribution. Hence, the increase in emissions in the area west of Zurich, north of BRM, and south of Basel looks reasonable.





**Figure 10.** Spatial distribution of a-posteriori emission for HFC-125 (a–d), $SF_6$ (e–h), and HFC-32 (i–l) for the high- and low-resolution inversions and the population-based (rows 1–2) and elevation-dependent a-priori (rows 3–4).





### 3.4 Emissions of HFC-32

Difluoromethane or HFC-32 is the fourth most emitted HFC in Switzerland with reported emissions of 57 Mg for 2020 and an increasing emission trend. HFC-32 is employed as a refrigerant for the same purposes as HFC-134a. Hence, we expect the spatial distribution of its emissions to be similar to HFC-134a. Its lifetime is only 5 years and its GWP is correspondingly low (705). Thus, it is a relatively low-risk choice among HFC refrigerants.

The BASE7 and BASE1 inversions for the period 2019–2021 yield a-posteriori annual emissions of $38\pm8$ and $50\pm8$ Mg yr$^{-1}$,
respectively (Figs. 10, 11, and (Fig. 9)). In Rust et al. (2022) the Swiss emissions estimate with PREL_SITRED7 inversion with population-based apriori emissions for HFC-32 was $44\pm12$,Mg for 2019–2020 (Figs. S4–S5).

Comparing (e) with (f) in Fig. 11 reveals again significant differences between the two model versions. While both increase the emissions in the canton of Valais, the Lausanne region, and around SOT, there is a significant difference for the rest of the Swiss Plateau, where the BASE7 inversion decreases the emissions, whereas the BASE1 inversion mostly
increases the emissions. In Fig. S6 the a-posteriori emission differences between the high- and low-resolution inversions are shown. These results are very similar to the results for HFC-125. The latter, together with the resemblance of the a-posteriori emissions between HFC-32, HFC-125, and HFC-134a for all model resolutions, verify our prior assumption that the emissions for these substances have similar sources.

In (g) and (h) in Figs. 10 and 11 the a-posteriori emissions and a-posteriori minus a-priori emission differences for HFC-
32 are depicted starting from an elevation-dependent a-priori (BASE_ED). Again the results are very similar to those of HFC-134a, and the a-posteriori emissions reconstruct again a population-based distribution. The hotspots of emissions in the cantons of Zurich and Aargau are reconstructed by the inversion, along with the hotspots in the Lausanne and Geneva regions. The total Swiss emission estimate for the BASE_ED1 inversion with the elevation-dependent a-priori is $47 \pm 5$ Mg yr$^{-1}$, while for the BASE_ED7 inversion is $35 \pm 4$ Mg yr$^{-1}$. The results for the inversions starting from a uniform
distribution by country (BASE_UNI), can be seen in the supplement (Figs. S18–S21). The statistical measures assessing the reliability and performance of the results are summarized in Table 3, confirming the generally improved performance of the high-resolution model at the receptor sites for all a-priori distributions.





**Figure 11.** Spatial distribution of a-posteriori minus a-priori emission difference for HFC-125 (a-d), HFC-32 (e-h), and SF$_6$ (i-l), for the high- and low-resolution inversions and the population-based (rows 1–2) and elevation-dependent a-priori (rows 3–4).



## 3.5 Emissions of $SF_6$

Sulfur hexafluoride or $SF_6$ is also an F-gas that is mainly used (80% of its emissions, Simmonds et al., 2020) in the electri-
cal power industry as a gaseous dielectric medium. Another utilization of $SF_6$ is in semiconductor manufacturing and as
inert gas for the casting of magnesium. The Swiss inventory value for $SF_6$ emissions was 6.7 and 6.0 Mg yr$^{-1}$ for 2019 and
2020, respectively. Although its emissions are lower compared to those of the HFCs presented before, its very large GWP
(23'500) makes $SF_6$ the fourth most important contributor of the F-gases to anthropogenic warming. Since $SF_6$ is used as
an electrical insulator by the electrical industry, its emissions may be concentrated on point sources and the choice of a
population-based a-priori may not be as obvious as with the HFCs.

The BASE7 and BASE1 inversions for the period 2019–2021, yield to a-posteriori annual emissions of $7.6\pm1.1$ and $9.0\pm1.1$ Mg yr$^{-1}$,
respectively (Figs. 10 and 11). Sensitivity tests using both uniform and population-based a-priori (Rust et al., 2022), showed
that the population-based a-priori is still the best option for this compound. Their Swiss emissions estimate for $SF_6$ – us-
ing only observations from BRM (PREL_SITRED7) and a population-based a-priori – was $9\pm2$ Mg yr$^{-1}$ for 2019–2020 (Figs.
S7–S9). These results are consistently larger than the inventory values. A potential cause of this discrepancy could be the
strong emission sources in southwest Germany (as indicated in our a-posteriori results; also compare (Simmonds et al.,
2020)), which potentially may have been mis-attributed to Switzerland as well.

Comparing (i) with (j) in Fig. 11 the distribution of the emissions for both model resolutions reveal very similar patterns
of increases and decreases of emissions compared to the a-priori. The difference of 15% in total emissions between the two
inversions comes from the areas of Lausanne, Geneva, and the areas north and north-west of BRM. The BASE1 inversion
increases the emissions more in these regions than the BASE7 inversion, Fig. S9.

In (k) and (l) in Figs. 10 and 11 the a-posteriori emissions and a-posteriori minus a-priori emission differences for $SF_6$
are summarized starting from an elevation-dependent a-priori (BASE_ED). The BASE_ED1 inversion does a better job of
reconstructing the hotspots of emissions north of BRM and around Lake Geneva. The total Swiss emission estimate for the
BASE_ED1 inversion is $7 \pm 1.2$ Mg yr$^{-1}$, while for the BASE_ED7 inversion is $5 \pm 1$ Mg yr$^{-1}$. The results for the BASE_UNI
inversions can be seen in the supplement (Figs. S18–S21), whereas the statistical measures assessing the reliability and
performance of the results are summarized in Table 3. In contrast to the other compounds, the spatial distribution of the
emissions for $SF_6$ –when a uniform a-priori is used– seems more reasonable and highlights the same emission regions as
the inversions with the population and the elevation-dependent a-priori.

## 4   Conclusions

This study highlights the importance of employing high-resolution meteorological fields and a dense observational net-
work to inversely estimate total emissions and their spatial distribution on the national scale (in this case a small country
in the order of 40,000 km$^2$). Although the computational cost increases with the increasing resolution of the NWP model
and the inclusion of additional observations, the differences in the national total emission estimates and their spatial dis-
tribution can be significant, as shown in this study for halocarbons.





Here, we used an analytical Bayesian inversion framework to minimize the observational and a-priori error, and hence, estimate the total Swiss halocarbon emissions and their spatial distribution. We first showed that including additional receptors in the neighboring countries does not affect the total Swiss emission estimates or their spatial distribution. If enough receptors are utilized to constrain the European emissions, then additional receptors – far from the region of inter-
est – do not lead to significant gains in that region itself.

We further investigated how variations in the inversion setup (e.g., inversion grid size, different spatial distribution of a-priori emissions, optimization of different covariance parameters during the maximum likelihood step, 3-hourly or 24-hourly observation aggregation periods, seasonal emission variability, additional receptors outside Switzerland) influence the final emission estimates, their uncertainty, and their spatial distribution. While most of these parameters do not show
any significant sensitivity to the final emission estimates and their uncertainty, the baseline uncertainty has a significant impact on the final inversion estimates. Future research should focus on comparing inversions using different baseline estimation methods (e.g., Vojta et al., 2022), since a reduced baseline uncertainty and an accurate baseline concentration will substantially benefit the inversion estimates.

In contrast to additional receptors far from the region of interest, the inclusion of additional measurements inside
Switzerland significantly amended both the total Swiss emission estimates for HFC-134a and partly their spatial distribution. Inversions with the high-resolution model with and without SOT differ by more than 10% in terms of total Swiss emissions: $307\pm48\,\mathrm{Mg\,yr^{-1}}$ for the inversion including only BRM versus $351\pm44\,\mathrm{Mg\,yr^{-1}}$ for the inversion including both BRM and SOT. These additional emissions in the inversion including two receptors on the Swiss Plateau are attributed to the big cities of Switzerland, thus, increasing the population-based signal of the emissions for this compound. This is com-
pletely rational since the inclusion of additional receptors, SOT, adds sensitivity to areas where BRM was not very sensitive.

Inversions solely assessing the effect of the transport model resolution on the Swiss emission estimates and their spatial distribution for HFC-134a showed significant differences across the two model resolutions. The total emission estimates discrepancy was close to 40% (i.e., $260\pm49\,\mathrm{Mg\,yr^{-1}}$ for the low-resolution inversion versus $351\pm45\,\mathrm{Mg\,yr^{-1}}$ for the high-resolution inversion), and the additional emissions of almost $100\,\mathrm{Mg\,yr^{-1}}$ were attributed to the big cities, industrial areas, and the traffic network. The high-resolution estimate is also closer to the bottom-up reported value of $415\,\mathrm{Mg\,yr^{-1}}$. The correlation of the a-posteriori emissions with the population and traffic $CO_2$ emissions is significantly higher for the inversion with the high-resolution model. Since we are confident that the emissions of this compound indeed follow a population-based distribution, the results clearly point to an improvement of the inversion when the high-resolution model is employed. This is also reflected in the correlation of the simulated values of HFC-134a with the observations at BRM and
SOT. Furthermore, when starting from an elevation-dependent a-priori the high-resolution model is able to reconstruct partially the emission hotspots and the population-based distribution while the low-resolution inversion does not recover these features to the same extent. Finally, the uncertainty of the total emission estimate is lower for the high-resolution inversion (12.8%) compared to the low-resolution inversion (18.8%). Inversions assessing the existence of an annual cycle in the emissions for HFC-134a revealed that for the high-resolution inversions there is a minimum of emissions during the
three winter months, DJF, and a maximum during the three summer months, JJA. Although the total emission estimates





and their spatial distribution do not change significantly when we consider seasonality, these inversions can depict the emission mechanism for substances with high uncertainty.

Inversions with the same setup for other F-gases (HFC-125, HFC-32, SF$_6$) lead to similar results as for HFC-134a. Results for these compounds have in common a significant difference in the total emissions (15–25%) and their spatial distribution

between the inversions with the high- and the low-resolution models. The high-resolution model inversions for all studied substances, except SF$_6$, converge closer to a population-based distribution and reveal additional emission hot spots. The regions north and north-west of BRM, west of Zurich, and south to the south-west of Basel are depicted as a high-emission area for both HFC-125 and HFC-32 by the high-resolution model inversions, while the low-resolution model inversions decrease the emissions compared to the a-priori in the same area. There is also a difference between the models near St.

Gallen, where again the high-resolution model increases the emissions, whereas the low-resolution model decreases the emissions. Both models agree on increased emissions in the canton of Valais, something highlighting the importance of additional observational sites (SOT). Inversions employing a different a-priori (elevation-dependent or uniform) for HFC-32 and HFC-125 perform less well and cannot depict the emission hot-spots. For SF$_6$ both models point to similar emission hot-spots, and this is the only substance for which we might expect the a-priori distribution to deviate strongly from the

population-based distribution. For this substance, the high-resolution inversions for the three different a-priori utilized can reproduce the same or similar emission regions, which is not true for the low-resolution inversions, in which when the uniform distribution is employed the low-resolution inversion fails and produces unrealistic results.

Our sensitivity inversions with a set of different a-priori highlight the importance of prior knowledge of the distribution of these emissions on a national level. If we start from an unrealistic distribution, the inversion will not be able to depict

the true emissions and their true spatial distribution. This can be seen in our inversions; the results for all HFCs and SF$_6$ are subpar when a uniform a-priori spatial distribution of the emissions is employed. The data themselves (likelihood function $P(\mathbf{y}|\mathbf{x})$) are not enough to drive the inversion to the true emission state when the a-priori spatial distribution of the emissions is unrealistic. Therefore, when the spatial distribution of the emissions on a national level is not known, a different set of spatial distributions should be tested.

Future work should focus on applying high-resolution inversions for other GHGs with biogenic sources and sinks (CO$_2$, CH$_4$), especially in countries with complex terrain for which the high-resolution meteorological fields should substantially improve the representation of atmospheric flow in the mountainous regions, and thus, the quantification and spatial attribution of the surface fluxes (Rotach et al., 2014). To correctly estimate GHG budgets of any compound using inverse modeling, we should both understand and mitigate the errors involved in the inverse framework. Transitioning to high-

resolution NWP models will drastically help to reduce the representation and model error and reconstruct the local atmospheric flows, while sensitivity inversions help characterize and understand the different errors involved in atmospheric inversions and how these affect the emission estimates and their spatial distribution.



*Code and data availability.* Continuous atmospheric halocarbon measurement data for the AGAGE stations are available in the following website (http://agage.mit.edu/data/agage-data, AGAGE, 2022). Measurement data for Tacolneston are available from the Centre for

Environmental Data Analysis (CEDA) archive (https://catalogue.ceda.ac.uk/uuid/a18f43456c364789aac726ed365e41d1, CEDA Archive, 2022). Beromünster measurement data are available from the Zenodo data repository (https://doi.org/10.5281/zenodo.5843548, Rust et al., 2021). Sottens measurement data will be made available in Zenodo data repository by the publication date of this study. Transport model results (FLEXPART footprints), inverse modelling code and, setups are available from the corresponding authors upon request. Input data used for running FLEXPART are proprietary data of MeteoSwiss and ECMWF, and can be obtained from these weather services

only.

*Author contributions.* IK carried out the transport model simulations and inverse modeling analysis of Swiss emissions. Measurements at Beromünster and Sottens were taken by DR with support from MKV. SO'D, DJ, KS, TS, JA, and MKV carried out observations at Mace Head, Tacolneston, Taunus Observatory, Monte Cimone and Jungfraujoch. The study was designed by IK, SH, and SR and supervised by SH. The manuscript was written by IK with support from SH and revisions from DR, DB, MKV, SR, LE, and all other co-authors.

*Competing interests.* The contact authors have declared that neither they nor their co-authors have any competing interests.

*Acknowledgements.* Funding from the Swiss National Science Foundation within the Project IHALOME (Innovation for Halocarbon Measurements and Emission validation) is acknowledged (SNSF, Project 20020_175921). Financial support for the measurements at Jungfraujoch has been provided by the Swiss National Programs HALCLIM and CLIMGAS-CH (FOEN), by the international Foundation for High Altitude Research Stations Jungfraujoch and Gornergrat (HFSJG), and by the Integrated Carbon Observation System Research

Infrastructure (ICOS-CH). We thank the personnel operating the Advanced Global Atmospheric Gases Experiment (AGAGE) measurement stations at Jungfraujoch, Tacolneston, ,Mace Head, Monte Cimone, and Taunus Observatory for conducting, evaluating, and providing the halocarbon measurement data. AGAGE operations are supported by the Upper Atmosphere Research Program of NASA, (grant nos. NAG5-12669, NNX07AE89G, NNX11AF17G, and NNX16AC98G to MIT; grant nos. NNX07AE87G, NNX07AF09G, NNX11AF15G, and NNX11AF16G to SIO) and the Department for Business, Energy, and Industrial Strategy (BEIS; grant no. TRN 1537/06/2018 to the Uni-

versity of Bristol for Mace Head and Tacolneston). We acknowledge MeteoSwiss, for providing meteorological observations and COSMO model analysis fields. FLEXPART-COSMO calculations were carried out at the Swiss National Supercomputing Centre (CSCS) under project grant s1091.





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
