# Peer review of "Impact of transport model resolution and a-priori assumptions on inverse modeling of Swiss F-gases emissions"

_Atmospheric Chemistry and Physics, 2022_

## Author Comment (AC1)

**Reply to comments by referee 1**

Referee comments in black.

Replies in blue.

*Suggested changes to text in italic green.*

The paper presents inverse modelling results using atmospheric transport models at varying spatial resolution. It is certainly of interest to the scientific community. In general the paper is well written, and I recommend publication after the following minor concerns have been addressed.

We would like to thank the referee for the detailed comments and overall favorable evaluation of our manuscript. We have addressed all questions in the following and modified the manuscript accordingly.

General Comments:

Regarding the sensitivity of inversion results to the assumed a priori emission distribution, it should be discussed a bit more why the inversion is not able to adjust and correct the spatial pattern. Which part is related to station density (coverage of the combined sensitivity) and which part is related to the lack of flexibility via the a priori uncertainty? It is a good suggestion to use different prior estimates with different spatial patterns, but then it needs to be ensured that those different estimates actually cover the range of possible distributions.

In the results of the manuscript a paragraph is already devoted to this comment:

*"Our sensitivity inversions with a set of different a-priori highlight the importance of prior knowledge of the distribution of these emissions on a national level. If we start from an unrealistic distribution, the inversion will not be able to depict the true emissions and their true spatial distribution. This can be seen in our inversions; the results for all HFCs and SF6 are subpar when a uniform a-priori spatial distribution of the emissions is employed. The data themselves (likelihood function P($y|x$)) are not enough to drive the inversion to the true emission state when the a-priori spatial distribution of the emissions is unrealistic. Therefore, when the spatial distribution of the emissions on a national level is not known, a different set of spatial distributions should be tested"*

Concerning to the a-priori uncertainty, we are using already too large uncertainties and as a result give room to the inversion to make big adjustments of emissions. The problem - I don't really like the word problem since this is how the mathematical framework works - lies in the Bayesian Inversion itself and in the nature of the a-priori. Most of the a-priori distributions used in Bayesian Inversions are informative (they express specific, definite information about a variable) and they drive the inversions by changing the shape of the cost function (Prior Probabilities). That's why one should either use an a-priori as uninformative as possible (which is difficult and it's a whole branch of mathematics trying to

produce uninformative priors) or should start from a a-priori distribution which is close to the reality, and they are confident with. The only way a bad a-priori can be overcome is by using a super-dense measurement network, which is also non-feasible.  The Bayesian inversion works by minimizing a cost function, and the cost is the difference of the true distribution, **x,** from the a-priori, **xb**, and the difference between the measurements**, y**, from the model value, **Mx** (Brasseur and Jacob 2017). In the Bayesian inversion, which is assumed to be a linear model, the cost function is convex - meaning that there is only one solution to the minimization problem – but the shape of the convex function, and hence the minimum, is highly influenced by the a-priori distribution. That's why - and since we don't have unlimited and perfect measurements – we should start from an a-priori which is close to the real distribution, or we should try different a-priori distributions, as we did, and discard the results which look unrealistic using robust arguments, something we also did in our paper. One of the main conclusions of the paper is that expert judgement is very important in these inversions since a bad a-priori could lead to an unrealistic a-posteriori distribution and to an overall bad model which could probably possess very good statistical metrics.

Specific comments

Lines 51-54: It was actually Lin et al (2003) who showed this for the first time.

Thank you, we changed the reference to (Lin et al. 2003; Seibert and Frank, 2004; Thomson and Wilson, 2012).

Lines 58-66: Errors in inversions and in transport have been discussed in the literature prior to 2018, please cite earlier studies.

The two cited papers were not meant to be exclusive. However, as the report by Bergamaschi et al. (2018) gives a broad overview of inverse modelling approaches for different greenhouse gases and scales, we thought it well suited here. We added "e.g." to the citation, to point out that these are only examples and we are aware of it. Furthermore, we added Lin and Gerbig 2005 and Lauvaux et al., 2009 to mention two other important examples.

Line 89: reference Bergamaschi et al., change year from 2017 to 2022

Thank you for spotting this mistake. Since the manuscript was published in its final version in the meantime, we updated the reference anyway.

Line 331: change "a-posterior" to "a-posteriori".

Done.

Line 345: ad a comma between the indices in the subscript (as in Eq. 12)

Done.

Line 349: a temporal correlation length of 0.01 days would only have an impact if the observations would be at a higher rate that about 1/hour. Is this the case? May be this should be mentioned clearly. It is not so clear that representation and model errors are uncorrelated between e.g. subsequent days or even hours, so this might need additional explanation or discussion.

It was already stated "... *that there is very low autocorrelation between daily average observations...*". To explain this more clearly we reworded as follows:

"... *that we assume almost zero auto-correlation (independency) between daily average observation/model errors. Although this may underestimate the true error correlation in some situations, in our experience it allows capturing pronounced pollution events more realistically in the a posteriori simulations.*"

Table 2: SEAS1 and SEAS2 have identical entries in the table, may be mention in a footnote to table 2 what the difference is.

The difference was described in the text (L455ff). However, we added a footnote to the table to make it more self-explanatory.

"*: Two different approaches for setting covariance parameters were used for the seasonal inversions. See text for details.*"

Line 458: please briefly explain the iterative approach here, what is iterated? Are simply posterior residuals used to inform on model-data mismatch error?

We added the following statement for additional explanation:

"*In the latter, the model-data error is first determined from the residuals of the a priori simulation, fitting a linear relationship to the residuals depending on a priori simulated concentrations. For subsequent iterations, the a posteriori residuals from the previous iteration are used instead. The method usually converges after 2-3 iterations.*"

Lines 484-486: What are typical scales for near- vs. far-field? This needs to be elaborated a bit more. To me it is unclear why in the far-field the diffusion should be depending on the size of the eddies, certainly at some distance the main cause of "diffusion" is the loss of correlation (or coherence) in the mean wind fields.

In my PhD thesis (https://www.research-collection.ethz.ch/handle/20.500.11850/578641 on page 22-23) the mean square displacement of a particle is given by equation 2.15 both for near- and far-field (screenshot is also attached below). These equations are derived by starting from the generic equation for the mean square displacement (see Csanady 1977 Turbulent diffusion in the environment for a thorough and analytic mathematical treatment) and assuming that the near field is the region with $t <<$ than the Lagrangian timescale and the far field where $t >>$ than the Lagrangian timescale. For times much lower than the Langrangian timescale, equation 2.17 (Langrangian autocorrelation function) has

a value of 1 and the integral of equation 2.16 below is equal to t leading to the upper branch of equation 2.15. This equation is independent of the Lagrangian timescales and the displacement of the particle depends only on time; hence the turbulence is independent of the size of eddies. There is a caveat here though. Since the near- and far-field dispersion is defined according to the Lagrangian timescales, for eddies in the Kolmogorov length scales the Lagrangian timescales will be very small, and as a result the near field will be very brief- order of milliseconds. For large eddies with large TLs the near-field can be in the order of a minute or so.  Apart from Csanady et al, the concept is very well described in the following "book" on pages 24-26: Turbulent Diffusion

> **Commented [hes1]:** Just give a proper reference here. Not a link. Add to the list of references.

The content of the sentence in the manuscript was enhanced and a citation was added.

"This happens because turbulent dispersion behaves differently in the near- and the far-field. In the near-field, dispersion approaches isotropy mainly at the large scales, meaning that the diffusion in the near-field is independent of the size of the eddies. During that phase, the size of the plume is much smaller than the larger turbulent eddies, and turbulence acts more like a mean transport mechanism (Csanady 1973)."

Line 494: remove comma after "overfitting"

Done.

Line 519: add "." at the end of the line sentence

Done.

Line 524: When I calculate the relative uncertainties, I get 12.5% and 18.8% for Base1 and Base7 respectively. Please correct

Done.

Figure 6 caption: e) and f) are missing

We changed the caption to Figure 6 to:

*A-posteriori minus a-priori emission differences for HFC-134a for the BASE inversion with the 7 km model (a, c, e) and the 1 km model (b, d, f) starting from a population-based a-priori (a, b), a spatially uniform a-priori (c, d), and an elevation-based a-priori (e, f).*

Line 542: Is it not expected that the reduced chi-square values are always close to 1 given that uncertainty covariance parameters are estimated using maximum likelihood?

Indeed, this is the expectation if the maximum likelihood search worked correctly. In addition, we force the state vector to a positive solution by adjusting the a-priori uncertainty in grid cells with negative a-priori results. This step follows the maximum likelihood search and, hence, a-priori covariance may have changed from what was used in the likelihood optimization.

Line 553: "rational state" not clear why that would not be rational, given the (wrong) prior, that is the solution one retrieves using a completely rational approach.

We agree and replaced the word 'rational' by 'realistic'.

Line 569: why is the uncertainty for the SEAS2 inversion results for JJA not given? It would also be interesting to discuss if the seasonality in retrieved fluxes is significant.

Thanks for spotting the missing uncertainty. We added the value to the text. It now reads 364 +/- 92 Mg yr$^{-1}$). We also added the following statement concerning the significance of the seasonality.

*"Given the relatively large a-posteriori uncertainties on seasonal emissions, summer and winter emissions are significantly different at the 95 % confidence level for the SEAS1 inversions but not for the SEAS2 inversions."*

Line 577: It would be interesting to see if taking into account seasonality improves the performance (e.g. using statistics as shown in Table 3). Not allowing for seasonal variations in combination with seasonal changes in transport patterns can be expected to result in larger model-data mismatch error.

Indeed, the model performance at the two Swiss Plateau sites was slightly increased when allowing for seasonal variations in the emissions. The performance gain in terms of RMSE was about 10 %. We added the following to the text and an additional table (similar to Table 3) to the supplement.

*"The a-posteriori model performance slightly increased for all seasonal inversion compared to the annual mean, population-based inversions (see supplement table xxx). For the RMSE, the performance increase was in the order of 10 %, but was less pronounced for the correlation coefficient. SEAS2 inversions achieved better performance statistics, but also revealed a chi2 index considerably larger than 1, indicating some degree of overfitting. "*

|          | X2   | DOF | R (BRM) | R (SOT) | RMSE (BRM) | RMSE (SOT) |
|----------|------|-----|---------|---------|------------|------------|
| C7 SEAS1 | 0.89 | 252 | 0.78    | 0.81    | 4.2        | 4.3        |
| C7 SEAS2 | 1.4  | 329 | 0.73    | 0.76    | 4.7        | 4.8        |
| C1 SEAS1 | 0.90 | 199 | 0.85    | 0.85    | 3.5        | 3.9        |
| C1 SEAS2 | 1.3  | 303 | 0.81    | 0.81    | 3.9        | 4.4        |

Line 597 / Table 3: it would be helpful to also have the posterior estimates and uncertainties included.

They are now included in Table 3.

Lines 681-685: It needs to be mentioned that the case of Switzerland is special given the orography. The manuscript has not shown similar impacts of resolution increase in domains with more benign orography.

Indeed, the situation in Switzerland is special because of the orography. However, other world regions deal with similar kinds of complexity that is not necessarily driven by orography alone, but by coastlines, dense population centers in otherwise sparsely populated environments, etc. We slightly reworded the sentence:

*This study highlights the importance of employing high-resolution meteorological fields and a dense observational network to inversely estimate national and sub-national emissions and their spatial distribution in regions with a complex emission distribution. In our case this is a small country of the order of 40,000 km$^2$ with complex orography. Other examples would be coastal regions or areas with skewed population/emission distributions and local flow patterns.*

Lines 694-695 "While … uncertainty," the sensitivity is the other way around (emission estimates being sensitive to parameters), please correct.

We corrected as follows:

*While the final emission estimates and their uncertainty did not show any significant sensitivity to most of these parameters, the baseline uncertainty had a significant impact on the final inversion estimate.*

Lines 736-737: May be reformulate "… which is not true for the low-resolution inversions, in which when the uniform distribution is employed the low-resolution inversion fails and produces unrealistic results." e.g. to "… which is not true for the low-resolution inversions, which fails and produces unrealistic results when using the uniform distribution."

We changed the sentence following your suggestion:

*"For this substance, the high-resolution inversions for the three different a-priori utilized can reproduce the same or similar emission regions, which is not true for the low-resolution inversion, which fails and produces unrealistic results when using the uniform distribution."*

---

## Author Comment (AC2)

**Reply to comments by referee 2**

Referee comments in black.

Replies in blue.

*Suggested changes to text in italic green.*

Katharopoulos et al presented a study that estimates posterior emissions of f-gases over Switzerland. In this study, the meteorological fields were driven using COSMO model with 7 (C7) and 1 km (C1), with an improved turbulent scheme for C1. Source-receptor relationships were obtained using FLEXPART model. Prior emissions were spatially and temporally distributed considering different assumptions, which resulted in a set of scenarios. In general, the results are consistent and in agreement with the literature. Authors claim that the uncertainty on baseline concentrations and spatial distribution of prior emissions have more impact on posterior estimates. However, the authors must explain and clarify some issues first. Also, there are some minor issues to be solved.

We would like to thank the referee for the detailed comments and overall favorable evaluation of our manuscript. We have addressed all questions in the following and modified the manuscript accordingly.

Line 55: explain why is harder with Eulerian models

We added the following text explaining why inverse modelling with Eulerian models may require additional efforts.

*"In contrast, deriving a source-receptor relationship from an Eulerian model either requires an adjoint version of the model, a finite-differences approach including multiple perturbed forward simulations or employing ensemble methods (e.g., Brasseur and Jacob, 2017), all of which come at a higher computational cost especially in situations with small amounts of observational data."*

Line 65: It is general knowledge of the complex terrain of Switzerland, however, it would be appreciated if the authors can provide some numbers.

We modified the section ending with the citation of Schmidli et al. (2018) and added more information on their findings and some general statements concerning the Swiss mountain topography.

*" … . Specifically for the typical Alpine topography it could be shown that valley wind systems of the major Alpine valleys like Rhone, Rhine, and Ticino, which have typical valley with of 4 to 8 km and, hence, cannot be sufficiently resolved at 7 km model resolution, are much better captured at 1 km model resolution (Schmidtli et al., 2018). Other smaller scale valleys remain too narrow to be properly resolved even at 1 km resolution. Another important feature of the Swiss Plateau is the flow channeling between Alps*

*and Jura mountains. With a distance between those two mountain chains of approximately 50 km this channeling is generally resolved at 7 km already."*

Furthermore, we come back to this point when discussion model performance in Section 3.2 following line 546, pointing out improvements in model performance in the C1 a-priori simulations, which we mostly attribute to improvements in flow representation in complex terrain.

*"A general performance improvement of FLEXPART-COSMO-1 versus FLEXPART-COSMO-7 can already be seen in the a-priori simulations (see supplement Table 2), where it is solely due to improvements in the transport description/flow in complex terrain, as emission distribution and baseline values were the same for both sets of simulations."*

Line 71: "Thus, a potential decline of fossil fuels will possibly see increased emissions from refrigerants" Can you explain the logic?

As we explained in the sentence above, tackling HFC emissions is often thought of as the low hanging fruit when it comes to reducing greenhouse gases since HFCs are currently much less connected to the energy sector, which is responsible for a large fraction of global anthropogenic $CO_2$ emissions. However, when moving away from burning fossil fuel in the heating sector by broadly introducing heat pumps, a large demand on refrigerants will result, some of which will eventually find their way into the atmosphere.. We slightly revised the sentence to make sure the connection with the heating sector becomes clearer.

*"Even if they do not play a significant role in the energy system now, they are increasingly used in heat pumps and air conditioners. Thus, a potential decline of fossil fuel use for heating may possibly see increased emissions from HFCs used as refrigerants."*

Line 190: How did you aggregate the observations?

The aggregates represent simple 24-hour mean values of all available observations within a given day. No further filtering by time-of-day was applied. We deemed this reasonable, since we recently showed that with the higher-resolution transport model we are able to capture the diurnal variations of trace species at the Beromünster tall tower reasonably well (Katharopoulos et al., 2022). As mentioned, we also ran sensitivity inversions with 3-hourly observations without seeing significant differences to 24-hourly observations. Hence, for the sake of computational costs we decided for 24-hour observations. Information on the aggregation method was added to the text.

*"To run our inversions, 24-hourly (3-hourly for sensitivity inversion) mean values were produced from the available observations of the above-mentioned sites."*

Line 197: Regarding REBS, can you mention other methods? Is it possible to use Thin Spline for instance?   Wood, S.N. (2003) Thin-plate regression splines. Journal of the Royal Statistical Society (B) 65(1):95-114.

We added an additional sentence on alternative methods after describing the details of the REBS approach. In general, differences between these purely observation-based approaches are not very large. Other approaches that use additional trajectory information may differ more strongly but obviously depend on a transport model. Thin-plate regression splines could be an additional option, however, the combination of a smoothed curve fit with a rejection/weighting scheme as provided by REBS seems to be the most promising approach.

*"Other statistical methods for baseline estimation have been applied to greenhouse gas observations (e.g., Thoning et al., 1989; El Yazidi et al., 2018) some of which use additional transport model information (trajectories or footprints) to select background sectors (O'Doherty et al., 2001). Differences between estimated background conditions are often small or limited to certain events or situations. There is no consensus which of these methods is most robust under all circumstances."*

Line 221: Is Flexpart capable of read and run with meteorological fields from ICON?

We are currently developing a version of FLEXPART for direct use of ICON output. This development is well advanced and we expect this version to become available to the ICON community towards the end of the year 2023. No changes to the text.

Line 244: A mention to Hysplit seems appropiate. Stein AF, Draxler RR, Rolph GD, Stunder BJ, Cohen MD, Ngan F. NOAA's HYSPLIT atmospheric transport and dispersion modeling system. Bulletin of the American Meteorological Society. 2015 Dec 1;96(12):2059-77.

Indeed, thank you for pointing out this lack. We added the reference to the revised manuscript as follows:

*"Nowadays, FLEXPART (Stohl et al., 2005; Pisso et al.,2019), and other LPDMs like NAME, HYSPLIT and STILT (Jones et al., 2007; Stein et al., 2015, Lin et al. 2003), are utilized for a large variety of tracer transport problems ..."*

Line 309: It is not very clear, but it seems that to reduce computation cost, cells with lower sensitivities are aggregated. However, higher prior fluxes in the same cells can produce convolved emissions which may be not neglected. Comment, please.

By aggregating to larger grid cells away from the observations, we tackle two problems. One is the reduced sensitivity, which would create many state vector elements with very little individual impact on the observations (although, as pointed out, this may be different for some high emission cells). Second and probably more important, is the smoothing of transport model errors, which grow with distance from the release location. By using larger grid cells at larger distances, transport errors tend to be smoothed out and the impact of individual mismatches (like missing a large distant source by a few kilometers in the simulated footprint) will be reduced. The following information was added:

*"The reduced resolution grid serves two purposes. On the one hand, it reduces the number of state vector elements, removing many elements with very little sensitivity. On the other hand, it helps to smooth out transport model errors, which tend to grow with distance from the point of observation."*

Line 410: A recent publication states that SF6 seems appropriate (Hu et aL., 2023)

Hu, L., Ottinger, D., Bogle, S., Montzka, S. A., DeCola, P. L., Dlugokencky, E., Andrews, A., Thoning, K., Sweeney, C., Dutton, G., Aepli, L., and Crotwell, A.: Declining, seasonal-varying emissions of sulfur hexafluoride from the United States, Atmos. Chem. Phys., 23, 1437–1448, https://doi.org/10.5194/acp-23-1437-2023, 2023.

Thanks for mentioning this article. We added the reference to the discussion of SF6 priors.

Line 446: Can you scale priori emissions according to temperature and season? Furthermore, for these emissions, using roads as a proxy seems more appropriate rather than the presented methods. Can you comment on that?

We did not scale the a priori emissions by some temperature proxy, but allowed the inversion to pick up any seasonal variability. Similarly for the spatial distribution of the emissions. We allowed the inversion to optimize it and discuss the result in terms of the spatial distribution of the road network, in section 3.2. No additional text added to the manuscript.

Line 453: Do you believe that scaling prior emissions with temperature by season would improve your posterior estimates? Please, comment.

From previous inversions on the same spatial scale but for CH4 and N2O we know that the inversion is very much able to pick up seasonal and monthly variability even if the a priori does not contain any variability. Hence, we did not think it necessary to prescribe such scaling for the HFCs either for which we expected smaller seasonal variations than for N2O emissions, which are strongly driven by soil processes and, hence, temperature and soil water. We added the following to the manuscript:

*From previous inverse modelling of Swiss CH4 and N2O emissions (Henne et al., 2016; FOEN, 2022) we know that the inversion was able to realistically pick up seasonal variability even if the a priori emissions did not include any variations in time.*

Line 477: A significant test, such as wilcox or mann-whiteney is required.

In line 476 we changed the use of "no significant spatial differences" to "no large spatial differences", since we agree that no statistical test confirms this observation.

Lines 487-491: Can you comment other methods?

We assume that this question aims at different methods for baseline optimization in general. We added the following text to the manuscript to cover such methods:

*"Estimating the baseline concentration purely from observations and optimizing it by site may not be the best solution to the baseline problem. Alternatively, baseline observations and transport model information can be used to reconstruct a spatially and temporally resolved baseline concentration at the domain boundaries from which, again with the transport model information, a baseline concentration for*

*each site and time can be sampled (e.g., Manning et al., 2021; Hu et al., 2023). Baseline concentrations at the domain boundary may then be included as part of the state vector.*"

Lines 580-583: Paragraph with only two phrases. Each paragraph must have at least three phrases.

In the revised manuscript, we combined these two very short paragraphs to one paragraph.

All figure uses too much blank space. It would be better remove longitudes and latitudes and use white space, leaving only latitudes at left side and longitudes at bottom side. In this way, the figures in the middle, would be without coordinates leaving more space to see the figures.

We thank the referee for this suggestion. We adjusted the amount of white space in figures 2, 8, 10, and 11, where this issue was most prominent.

**Additional References**

Hu, L., Ottinger, D., Bogle, S., Montzka, S. A., DeCola, P. L., Dlugokencky, E., Andrews, A., Thoning, K., Sweeney, C., Dutton, G., Aepli, L., and Crotwell, A.: Declining, seasonal-varying emissions of sulfur hexafluoride from the United States, Atmos. Chem. Phys., 23, 1437-1448, doi: https://doi.org/10.5194/acp-23-1437-2023, 2023.

El Yazidi, A., Ramonet, M., Ciais, P., Broquet, G., Pison, I., Abbaris, A., Brunner, D., Conil, S., Delmotte, M., Gheusi, F., Guerin, F., Hazan, L., Kachroudi, N., Kouvarakis, G., Mihalopoulos, N., Rivier, L., and Serça, D.: Identification of spikes associated with local sources in continuous time series of atmospheric CO, CO2 and CH4, Atmos. Meas. Tech., 11, 1599-1614, doi: https://doi.org/10.5194/amt-11-1599-2018, 2018.

Manning, A. J., Redington, A. L., Say, D., O'Doherty, S., Young, D., Simmonds, P. G., Vollmer, M. K., Mühle, J., Arduini, J., Spain, G., Wisher, A., Maione, M., Schuck, T. J., Stanley, K., Reimann, S., Engel, A., Krummel, P. B., Fraser, P. J., Harth, C. M., Salameh, P. K., Weiss, R. F., Gluckman, R., Brown, P. N., Watterson, J. D., and Arnold, T.: Evidence of a recent decline in UK emissions of hydrofluorocarbons determined by the InTEM inverse model and atmospheric measurements, Atmos. Chem. Phys., 21, 12739-12755, doi: 10.5194/acp-21-12739-2021, 2021.

O'Doherty, S., Simmonds, P. G., Cunnold, D. M., Wang, H. J., Sturrock, G. A., Fraser, P. J., Ryall, D., Derwent, R. G., Weiss, R. F., Salameh, P., Miller, B. R., and Prinn, R. G.: In situ chloroform measurements at Advanced Global Atmospheric Gases Experiment atmospheric research stations from 1994 to 1998, J. Geophys. Res., 106, 20429-20444, doi: 10.1029/2000JD900792, 2001.

Thoning, K., Tans, P., and Komhyr, W.: Atmospheric Carbon Dioxide at Mauna Loa Observatory 2. Analysis of the NOAA GMCC Data, 1974–1985, J. Geophys. Res., 94, 8549-8565, doi, 1989.